# TRAINING LANGUAGE MODELS TO SUMMARIZE NARRATIVES IMPROVES BRAIN ALIGNMENT

**Khai Loong Aw**[1,2]    **Mariya Toneva**[1]
[1]Max Planck Institute for Software Systems    [2]Singapore Management University
`klaw.2020@smu.edu.sg  mtoneva@mpi-sws.org`

## ABSTRACT

Building systems that achieve a deeper understanding of language is one of the central goals of natural language processing (NLP). Towards this goal, recent works have begun to train language models on narrative datasets which require extracting the most critical information by integrating across long contexts. However, it is still an open question whether these models are learning a deeper understanding of the text, or if the models are simply learning a heuristic to complete the task. This work investigates this further by turning to the one language processing system that truly understands complex language: the human brain. We show that training language models for deeper narrative understanding results in richer representations that have improved alignment to human brain activity. We further find that the improvements in brain alignment are larger for character names than for other discourse features, which indicates that these models are learning important narrative elements. Taken together, these results suggest that this type of training can indeed lead to deeper language understanding. These findings have consequences both for cognitive neuroscience by revealing some of the significant factors behind brain-NLP alignment, and for NLP by highlighting that understanding of long-range context can be improved beyond language modeling.

## 1 INTRODUCTION

Language models trained to predict the next word over millions of text documents have led to large improvements on a range of benchmarks in natural language processing (NLP). However, researchers have shown that NLP models may rely on shallow heuristics to perform these tasks rather than on a deeper language understanding (McCoy et al., 2019; Min et al., 2020; Linzen, 2020). To build systems with deeper language understanding, recent work proposed to train language models on narrative datasets which require extracting the most critical information by integrating across long contexts (Kryscinski et al., 2021; Sang et al., 2022; Kočiský et al., 2018). Does this approach truly lead to a deeper understanding of language? We investigate this by turning to the one language processing system that truly understands complex language: the human brain.

Prior work has used human brain recordings to interpret representations of pretrained language models (LMs) (Søgaard, 2016; Toneva & Wehbe, 2019; Abdou et al., 2021). They evaluate how well representations obtained from a language model can predict representations sampled from the human brain during language comprehension via a brain imaging device, such as functional magnetic resonance imaging (fMRI). If this prediction performance, also known as brain alignment, is determined to be significant via a statistical test, then the LM and specific brain location or timepoint are thought to significantly align in their representations of language. Using these methods, researchers have shown that pretrained language models significantly predict large parts of the brain regions that are thought to underlie language comprehension (Wehbe et al., 2014b; Jain & Huth, 2018; Toneva & Wehbe, 2019; Caucheteux & King, 2022; Schrimpf et al., 2021; Goldstein et al., 2022).

In this work, we draw insights from the human brain to study whether language models are truly learning deeper language understanding. We analyze the effect of training LMs for narrative summarization on their alignment with fMRI recordings of human subjects reading a book chapter.

---

Code available at `https://github.com/awwkl/brain_language_summarization`.

We specifically investigate 4 pretrained language models (i.e., "base models") and 4 corresponding models obtained by training the base models on the BookSum dataset (Kryscinski et al., 2021) to improve the base language model's narrative understanding (i.e., "booksum models"). The Book-Sum dataset was selected because it is a summarization dataset that requires understanding complex interactions across long narratives. The 4 models were selected because their architectures were designed to integrate information across long contexts. We evaluate the alignment of the base and booksum models with fMRI recordings of 8 participants reading a chapter of a popular book word-by-word, made publicly available by Wehbe et al. (2014a). This dataset was chosen because it is one of the largest datasets of participants processing a narrative story (5176 words which corresponds to approximately 1300 samples of fMRI recordings per participant).

Our main contributions are as follows:

1. In Section 4, we show that training language models for deeper narrative understanding improves alignment to human brain activity. Also, when increasing the number of words fed to the models, up to 500 words, brain alignment increases. Lastly, for each model, we identify the layers where these improvements in brain alignment occur.

2. In Section 5, we show that improved brain alignment in Section 4 is not due to improved language modeling (LM) ability, a possible confounding factor. By disentangling LM ability's contribution to brain alignment, we present evidence that BookSum-trained models develop deeper language understanding.

3. In Section 6, we present a simple interpretability approach to study what brain-relevant information is gained by language models after training for deeper language understanding. Our results reveal that these models are learning richer representations across all tested discourse features (Characters, Emotions, Motions). Furthermore, they learn more about Characters than Emotions and Motions. This indicates that discourse features are a promising dimension to study brain alignment and deep language understanding.

Combined, our contributions from Sections 4, 5, and 6 present evidence that models trained to summarize narratives indeed develop deeper language understanding. The first reason is that improved alignment to human brains' deep understanding of characters, emotions and motions suggests the model has developed richer representations of these entities and concepts. Second, we focus on brain regions suggested by previous research to underlie language comprehension in humans. Hence, improved brain alignment is not spuriously related to non-language brain activities. Third, we show that brain alignment improves only when we provide longer input contexts (20 to 1000 words) to the LMs, which may be important for deep contextual understanding.

## 2 RELATED WORK ON BRAINS AND LANGUAGE

Our work relates to a growing body of research on disentangling the contributions of different types of information towards brain alignment. Toneva et al. (2022a) present an approach to disentangle supra-word meaning from lexical meaning in language models (LMs) and show that supra-word meaning is predictive of fMRI recordings in two language regions (anterior and posterior temporal lobes). Caucheteux et al. (2021) and Reddy & Wehbe (2021) disentangle alignment due to syntactic and semantic processing. Toneva et al. (2022b) examine if LM representations align with different language processing regions in different ways. Also, researchers have suggested that one contributor to the alignment is the LM's ability to predict the next word, with a positive relationship between next-word prediction ability and brain alignment across LMs (Schrimpf et al., 2021; Goldstein et al., 2022). However, more recent work shows that no simple relationship exists, and language modeling loss is not a perfect predictor of brain alignment (Pasquiou et al., 2022; Antonello & Huth, 2022). Merlin & Toneva (2022) introduced perturbations to disentangle alignment due to next word prediction and semantic knowledge. Our work contributes to this research area by disentangling the contributions of LM ability from deep language understanding towards brain-NLP alignment.

Some works investigated the alignment of fine-tuned language models with brain recordings. Schwartz et al. (2019) finetuned a pretrained BERT to predict fMRI and MEG recordings of people reading a book chapter, leading to improved prediction of previously unseen brain recordings, specifically in regions known to support language processing. However, it is not clear what information had been induced in the fine-tuned BERT model that contributed to improved brain alignment.

Oota et al. (2022) investigate brain alignment of BERT fine-tuned on a variety of NLP tasks, such as co-reference resolution and question-answering. They show the best aligning task-tuned BERT depends on whether the brain recordings correspond to reading or listening to the language stimuli. Our approach complements these works, but focuses on LMs trained to summarize narratives.

Additionally, prior work on LM brain alignment use LMs not specifically designed to integrate rich info across long contexts. Thus, they investigated relatively short contexts and observed that brain alignment worsened with increased sequence length for many models (Toneva & Wehbe, 2019). Differently, our work uses LMs trained to extract crucial info from long contexts. Brain alignment remains high even with sequence lengths up to 500 words, a finding that adds to prior works.

## 3 METHODS

### 3.1 BRAIN REPRESENTATIONS

We use a publicly available brain dataset (Wehbe et al., 2014a) consisting of fMRI recordings of 8 participants reading chapter 9 of the book *Harry Potter and the Sorceror's Stone* (Rowling et al., 1998). The 5176 words in the chapter were presented to participants at a fixed interval of 0.5 seconds, one word at a time. The chapter was divided into four runs of approximately equal length, and participants were allowed a short break at the end of each run. We use this specific dataset for two main reasons: (1) the story is situated within a rich narrative world with characters, emotions and relationships, and (2) it is one of the largest publicly available datasets in terms of samples per participant, which improves the reliability of the brain alignment estimate.

The fMRI recordings were sampled at a consistent interval, or repetition time (TR), of 2 seconds. At each TR, the activity level was recorded in each volume pixel (i.e., voxel) in the participant's brain. Hence, the brain representation for each participant is a matrix $Y_i \in \mathbb{R}^{n \times v_i}$, where $n$ is the number of TRs and $v_i$ is the number of voxels in the brain of participant $i$. The number of voxels differs for each human subject, ranging from $25,000$ to $31,000$.

### 3.2 NLP REPRESENTATIONS

We extract NLP representations from a total of 8 models. We refer to 4 models as "base models": BART-base, LED-base, BigBird-base and LongT5-base. We refer to the remaining 4 models as "booksum models": BART-booksum, LED-booksum, BigBird-booksum and LongT5-booksum.

**Base models.** Our 4 "base models" were trained with a language modeling (LM) objective. We use pretrained models from Hugging Face (Wolf et al., 2020). We provide more details in Appendix A.

- BART (Lewis et al., 2019) uses a standard Transformer-based sequence-to-sequence architecture with 12 layers and 768-dimensional representations. It can process sequence lengths up to 1024 tokens. BART was pretrained by corrupting documents and learning to reconstruct the original text, on a combination of books (narratives) and Wikipedia data.

- BigBird (Zaheer et al., 2020) is a Transformer-based model with 32 layers and 1024-dimensional representations. It uses a sparse-attention mechanism that allows it to process sequence lengths up to 4096 tokens. We use the version of BigBird that was pretrained with Masked Language Modeling on books and summarization of U.S. patent documents (BigPatent dataset).

- Longformer Encoder Decoder (LED) (Beltagy et al., 2020) has an encoder-decoder Transformer architecture with 12 layers and 768-dimensional representations. It uses an efficient local+global attention mechanism based on the Longformer, and can process up to 16384 tokens. LED was pretrained with the same procedure as BART above.

- LongT5 (Guo et al., 2021) is an encoder-decoder Transformer with 12 layers and 768-dimensional representations. It can handle long input sequences up to 16384 tokens. We use the version of LongT5 with Transient Global Attention (TGlobal). LongT5 was pretrained by masking and generating sentences on the C4 dataset, a collection of roughly 750 GB of English texts sourced from the public Common Crawl open repository.

**Booksum models.** Our 4 "booksum models" are first initialized from one of the "base models". All their parameters are then trained using the BookSum dataset (Kryscinski et al., 2021). We describe

the BookSum dataset in the paragraph below. We chose base and booksum models that would be useful for investigating our hypothesis. (1) Their architectures are designed to take advantage of long contexts, which may be useful for deep understanding. (2) The booksum models have good performance on BookSum, and the base models have good performance on other long-context benchmarks. (3) The base models, except LongT5, were pretrained with language modeling on large narrative datasets, which reduces the differences in training domains between the base and booksum models, to allow for more fair comparisons. (4) Our models are sufficiently varied, and we study 4 pairs of models in total, which improves reliability of our results.

**BookSum dataset for long-range summarization.** The BookSum dataset (Kryscinski et al., 2021) is a challenging dataset for summarizing long narratives. For training, the inputs are chapters from books and the outputs are human-written summaries. The input documents are relatively long, with an average of more than 5000 words, and contain long-range causal and temporal dependencies. The ground-truth output summaries have an average of roughly 500 words. To generate these summaries, NLP models must capture the most critical information by understanding complex interactions across long-range contexts. Some examples from the BookSum dataset are provided in Appendix B. Furthermore, the dataset belongs to the narrative domain, which captures rich discourse structures for characters, emotions, and relationships, which is also the case for the Harry Potter stimulus dataset that we use to evaluate brain alignment. The BookSum Dataset also does not include any books from the Harry Potter series, thus preventing train-test overlap.

**Extracting NLP representations.** Our goal is to extract NLP representations for each of the 5176 words in our Harry Potter fMRI dataset. For each word, we feed the word and its preceding context to a model, and extract its outputs at every layer for the word. We extract NLP representations for all layers of all 8 NLP models and vary the input sequence lengths between 1 to 1000 words. Each unique combination of (model, layer, input length) produces a different set of NLP representations, $X_{l,s}^m \in \mathbb{R}^{5176 \times d}$, where $d$ is the embedding size for model $m$, layer $l$, sequence length $s$.

### 3.3 ALIGNING BRAIN AND NLP REPRESENTATIONS.

We follow a general alignment approach previously used in several works (Jain & Huth, 2018; Toneva & Wehbe, 2019; Schrimpf et al., 2021). This approach learns a function to predict the fMRI recordings at every voxel of each participant $i$ using the NLP representations that correspond to the same text read by the participant. Similarly to previous work, we parameterize this function as a linear function, regularized using the ridge penalty. We train this function in a cross-validated way and test its performance on the data that was heldout during training. We select the ridge parameter via nested cross-validation. Since the fMRI data was collected in 4 runs of approximately equal length, we use 4-fold cross-validation where each fold corresponds to holding out one run of fMRI data for testing. We also remove 10TRs of fMRI data and the corresponding NLP representations from the ends of each test run to avoid possible train-test overlap. We provide other details about the alignment approach in Appendix C.

**Evaluate brain alignment.** We evaluate brain alignment using two evaluation metrics, computed between the predictions of heldout fMRI recordings $\hat{Y}_i \in \mathbb{R}^{n \times v_i}$ and the true corresponding fMRI data $Y_i \in \mathbb{R}^{n \times v_i}$. Both evaluation metrics are used by prior works and each has its own advantages.

- **20v20 classification accuracy.** This metric evaluates the fMRI predictions by using them in a classification task on held-out data in the cross-validation setting. The classification task is to predict which of two sets of words was being read, based on the corresponding NLP representations of these words (Mitchell et al., 2008; Wehbe et al., 2014b). The sets consist of 20 consecutive TRs and the classification is repeated 1000 times within each fold. The final 20v20 accuracy is the average across the folds. This metric was proposed to boost the signal-to-noise ratio in estimating the brain alignment for single-trial data. The Harry Potter data is entirely single-trial (i.e., the participants read the book chapter only once and so each word appears only once in its context) and increasing the number of TRs that are classified together up to 20 improves the classification accuracy for this dataset (Wehbe et al., 2014a;b).

- **Pearson correlation.** This metric evaluates the similarity between the fMRI predictions and the corresponding true fMRI data by computing the Pearson correlation between $\hat{y}_{i,j}$ and $y_{i,j}$ for participant $i$ and each voxel $j \in \{1, \dots v_i\}$. This produces a vector $\in \mathbb{R}^{v_i}$. This metric can be

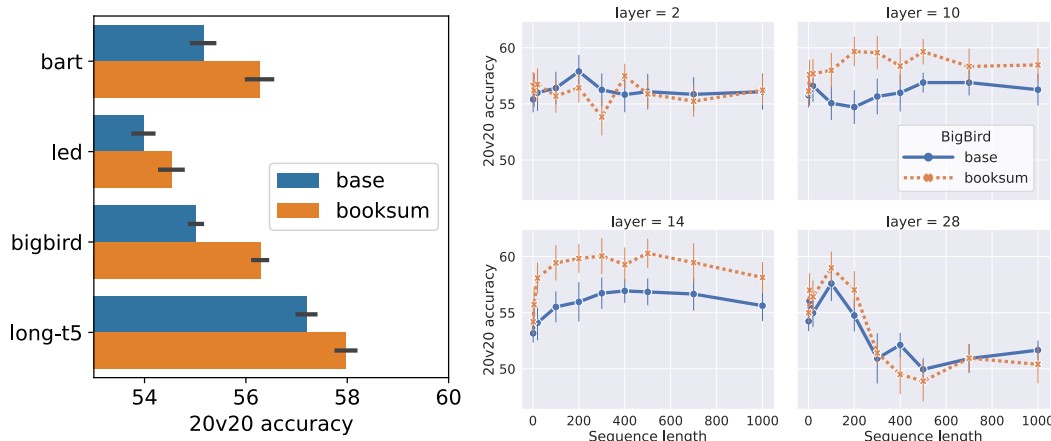

Figure 1: *Left.* Average 20v20 brain alignment for all 4 pairs of base and booksum models. Averages were computed over 8 layers for each model, sequence lengths 1 to 1000, and all 8 participants. See Appendix G for similar Pearson correlation plots. *Right.* Brain alignment for BigBird-base and BigBird-booksum, across representative layers and sequence lengths. Each data point is an average across 8 participants. Training for deeper understanding improves brain alignment significantly in early to middle layers (layers 6-14 out of 32) (paired t-test, FDR corrected for multiple comparisons at significance level 0.05). Results for the remaining models (BART, LED, LongT5) also improve significantly, presented in Appendix H. These results suggest that long-range and deep understanding is an important contributor to brain alignment.

computed efficiently and is widely used in cognitive neuroscience (Huth et al., 2016; Jain & Huth, 2018; Caucheteux et al., 2021; Schrimpf et al., 2021; Goldstein et al., 2022).

## 4    TRAINING NLP MODELS FOR DEEP NARRATIVE UNDERSTANDING

Do NLP models trained for deeper understanding of long narratives have improved brain alignment over those trained for language modeling (LM)? To investigate this, we evaluate the 20v20 brain alignment scores of 4 booksum models (deeper understanding) against 4 base models (LM), as described in Section 3. We use all 8 NLP models, layers 1 to 12, sequence lengths 1 to 1000, and all 8 human subjects. In Figure 1, we present average results for all 4 pairs of models, as well as results for a representative set of layers for one of the models. Other results are in Appendix G and H.

**Booksum models are significantly better aligned to brain representations.** We suggest that this is due to deeper understanding of text, because of two key facts. (1) Booksum models were trained on the BookSum dataset, which is designed with challenging tasks to learn deeper narrative understanding. It requires understanding complex interactions across long-range contexts and generating summaries that capture the most critical information. (2) Furthermore, these models have architectures designed to integrate information across long context lengths. This allowed them to develop deeper understanding when trained on BookSum.

Could the improved brain alignment of booksum models be due to other confounding variables and not deeper language understanding? We tried to eliminate possible confounding variables in the following ways: (1) Both base and booksum models were trained with similar data: large narrative datasets. Hence, improved brain alignment is not simply because booksum models were trained on data from a more similar domain to the brain alignment dataset (Harry Potter). We also verified this by comparing booksum models against base models trained on BookSum with a language modeling objective in Appendix M. (2) The booksum models and their corresponding base versions support identical sequence lengths, and were also trained with similar input lengths: all the base models were trained with long input sequences between 1024 and 4096 tokens. Hence, improved brain alignment is not because booksum models were trained on longer input lengths. (3) In Section 5, we show that training NLP models for deep understanding improves brain alignment, but does not improve LM ability. Hence, improved brain alignment is not because booksum models developed better

language modeling ability through BookSum-training. (4) In Appendix L, we show that the brain alignment of BART trained to summarize CNN news articles (BART-cnn) is not substantially higher than BART-base. Hence, the improved brain alignment is not simply due to the summarization training objective. Therefore, we argue that improved brain alignment for booksum models is likely not due to these confounding variables.

**Brain alignment across models peaks around context length of** 500 **words.** We further studied how brain alignment is affected by input context length to the models, which we varied from 1 to 1000 words. If brain alignment continues to increase as we increase context length, it would suggest that these models can indeed maintain information across long contexts. Prior work that used earlier language models such as BERT and Transformer-XL found that brain alignment peaked around 50 words (Toneva & Wehbe, 2019). We find a 10-fold increase in the sequence length that results in the peak of brain alignment, across both the base and booksum models. Because this peak is consistent for both types of models, we attribute the 10-fold increase to the fact that the models we evaluate in this work are specifically designed to integrate longer context than previous models. However, we do not observe an additional benefit beyond 500 words of context, even though the models we analyze are supposed to incorporate more than 500 words of context (up to 16384 tokens for LED and LongT5).

**Brain alignment improves only for longer input contexts (20 to 1000 words), but not short contexts (1 to 5 words).** Importantly, this suggests: (1) improved brain alignment is likely due to deeper language understanding (which benefits from longer contexts), and (2) these models trained for deeper understanding have developed better ability to make use of long contextual information.

**Booksum finetuning improves brain alignment for different layers across models.** Finetuning has been shown to result in greater changes in deeper layers than earlier layers of models (Mosbach et al., 2020). Therefore one might expect that changes in brain alignment due to finetuning may emerge primarily in the deeper layers. However, we find that the brain alignment is significantly improved at different layer depths across the 4 models that we test. BigBird and LED improve their alignment with the brain significantly in early to mid layers (layers 6-14 out of 32 in BigBird, and layers 2-8 out of 12 in LED), whereas LongT5 shows significant increases only in the deep layers (layers 8-12 out of 12). BART shows a consistent significant difference throughout most of its layers (layers 2-11 out of 12). These results suggest that the brain-relevant changes may be distributed differently across different models.

## 5 RELATIONSHIP BETWEEN BRAIN ALIGNMENT, LANGUAGE MODELING AND DEEP UNDERSTANDING

Why do language models trained on BookSum have improved brain alignment, as our results show in Section 4? Is it due to the models developing deeper language understanding, or are there other confounding factors? One possible confounding factor is language modeling (LM) ability, as some works suggest that LM ability contributes to brain-NLP alignment (Schrimpf et al., 2021; Caucheteux & King, 2022; Goldstein et al., 2022). This raises the possibility that booksum models have greater brain alignment simply because they developed better language modeling ability through BookSum-training. We conduct experiments to further investigate this possibility.

Our goal is to show that improved brain alignment is related to deeper understanding, rather than improved LM ability. We show this by demonstrating that training LM models for deep understanding (booksum models) improves brain alignment, but simultaneously does not improve LM ability for 3 of the 4 models.

We evaluate the LM ability of all 8 NLP models (4 base, 4 booksum) on the text corresponding to the same Harry Potter chapter which was used to obtain the brain recordings. First, we split our Harry Potter text dataset into a train and test set (75% and 25% of the text respectively). Next, for each model, we freeze all layers, add a LM head, and train only the head to predict masked words in the train set. Finally, we evaluate each model's cross-entropy loss (LM ability) on the heldout test set. We froze all model weights (other than LM head) to ensure that the representations are preserved, identical to the ones used to evaluate brain alignment. We used the Harry Potter text dataset to ensure that the same dataset is used for measuring both brain alignment and LM ability, to

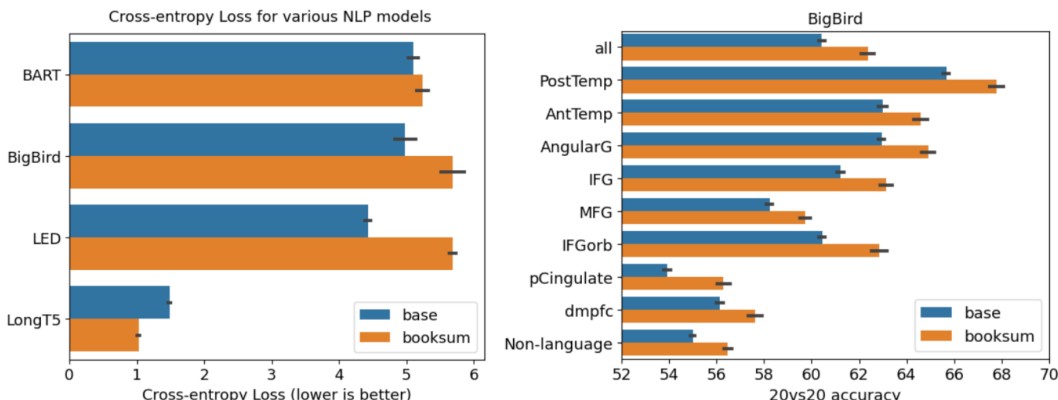

Figure 2: *Left.* Language modeling (LM) loss (lower is better) on Harry Potter dataset. Booksum model has worse LM ability than base model for BigBird and LED, better for LongT5, and no significant difference for BART. Hence, while all 4 booksum models have better brain alignment, most do not have better LM ability. Thus, improved brain alignment in booksum models (Figure 1) is due to deeper understanding, and not language modeling ability. *Right.* 20v20 accuracy is significantly higher for the model trained for deeper understanding (booksum) than LM model (base), across all brain language regions (paired t-test, FDR corrected for multiple comparisons). Averages were computed over 8 layers for each model, sequence lengths 1 to 1000, and all 8 participants. Similar results were obtained for all other models (BART, LED, LongT5) (Appendix I). This suggests that the function of deep understanding is not localized to any one particular region in the brain, but is instead distributed across language regions.

allow for a consistent comparison. We perform 4-fold cross-validation to improve reliability, where each fold corresponds to consecutive 25% of the text. The results are presented in Figure 2.

**Improved brain alignment is not due to better language modeling (LM) ability.** For BigBird and LED, the booksum model achieves significantly higher brain alignment but significantly worse LM performance than the base model. For BART, the booksum model achieves greater brain alignment but does not significantly differ in LM performance. For LongT5, the booksum model has both higher brain and LM performance. Hence, while all 4 booksum models have significantly better brain alignment, the majority of them do not have a better LM performance. This shows that the improved brain alignment for at least 3 of the 4 models is due to deeper understanding and not to their ability to predict the next word.

How does brain alignment differ across brain regions? We specifically investigate alignment within the brain regions in the language network Fedorenko et al. (2010); Fedorenko & Thompson-Schill (2014), which are thought to underlie language comprehension, and two additional language regions Binder et al. (2009), which are thought to support word semantics. For each brain region, we compare the average brain alignment for booksum versus base models. (Figure 2 and Appendix I).

**Across all considered regions, brain alignment is significantly greater for booksum models.** This suggests that brain-relevant information imparted by training towards deep understanding is not localized to any one particular brain region, but instead distributed across language regions. Furthermore, this shows that models with significantly better language modeling performance (base ≫ booksum for BigBird and LED) do not achieve better alignment in any language region. This further supports the need to complement language modeling with training for deep understanding.

## 6    INTERPRETING BRAIN ALIGNMENT ACROSS DISCOURSE FEATURES

What are NLP models learning when trained to summarize narratives? Are they learning a deeper understanding of important discourse features in the text, such as Characters, Emotions, and Motions? These discourse features have representations that go beyond the meaning of their individual words. Characters develop over the course of the entire book through their thoughts, actions, and interactions with other characters. Emotions depend on who is experiencing them, what situation

**Algorithm 1** Interpretability approach to compare brain alignment across discourse features

1: **Input:** fMRI predictions, fMRI activity
2: **for all** discourse features **do**
3:     label words with discourse feature
4:     map labeled words to fMRI TRs
5:     extract NLP predictions for discourse TRs
6:     extract brain activity for discourse TRs
7:     compute Pearson corr. for discourse TRs
8:     **Result:** Pearson corr. for discourse feature
9: **end for**
10: **Result:** Pearson corr. for all discourse features

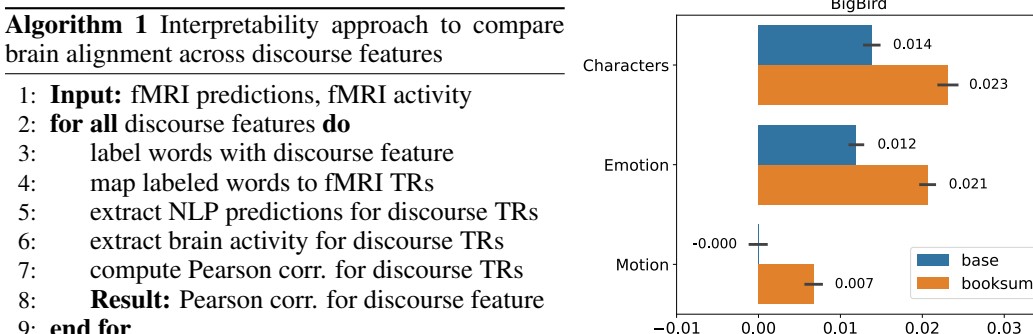

Figure 3: *Left.* Interpretability approach to compare Pearson correlation brain alignment for fMRI samples corresponding to various discourse features. *Right.* Pearson correlation averages for three discourse features. Averages were computed over 8 layers for each model, sequence lengths 20 to 500, and all 8 participants. NLP models have greater brain alignment for Characters than other discourse features. When trained to summarize narratives, the models improve their brain alignment significantly for all discourse features (paired t-test, FDR corrected for multiple comparisons). However, it improves more for Characters than other discourse features. Note that the average correlations shown here are low in magnitude as they include a large number of brain voxels that may not be significantly involved in brain-NLP alignment or language processing, as well as many layers and sequence lengths.

they are in, and why. Motions (actions) are situated in a scene, contextualized by the actor, target, rationale of the action, and other people and objects in the environment. Hence, richer representations of these discourse features suggest a deeper linguistic understanding of the text.

**Our simple interpretability approach to compare brain alignment across discourse features.** For each discourse feature, we compute an average Pearson correlation for booksum models, and another for base models. Different discourse features may be present at different frequency in the text. Hence, to enable fair comparisons, we use the same number of samples to compute the correlation for each discourse feature. We present the results and the approach's pseudocode in Figure 3. Averages were computed over 8 layers for each of the 8 models, sequence lengths 20 to 500, and all 8 participants. We provide more details in Appendix D.

**NLP models align more with the brain for Characters than other discourse features, and also improve the most for Characters when training to summarize narratives.** For Characters, Pearson correlation averaged across all voxels improves significantly more than other discourse features for most models ((paired t-test, FDR corrected for multiple comparisons; Appendix J. Note that the correlations reported in Figure 3 are low in magnitude because they are an average that includes a large number of brain voxels that may not be significantly involved in brain-NLP alignment or language comprehension, as well as over many layers and sequence lengths. Next, we compare the brain plots in rows D and E of Figure 4. Character TRs improve in more voxels (orange regions) than the random TR sample. It is also surprising that booksum-training actually results in significantly worse alignment for some brain voxels (blue regions). One possible reason is that these specific regions may participate in prediction of future words. As we saw in Section 5, the language modeling performance of BigBird declined due to BookSum-training.

**Discourse features are a promising dimension to study brain alignment.** Training LMs to summarize narratives improved brain alignment (richer representations) for all discourse features. Our work contributes towards understanding what LMs learn when trained for deeper understanding. We show that brain alignment is higher for Characters than other discourse features. We hypothesize that, for Characters, language models integrate richer info from surrounding context, producing representations with greater alignment to brain representations. On the other hand, LM representations for other discourse features may be more shallow and less context-dependent. In future, we hope to dive deep into investigating this hypothesis, ultimately generating insights for a better understanding of NLP representations. Also, future work can explore additional discourse features.

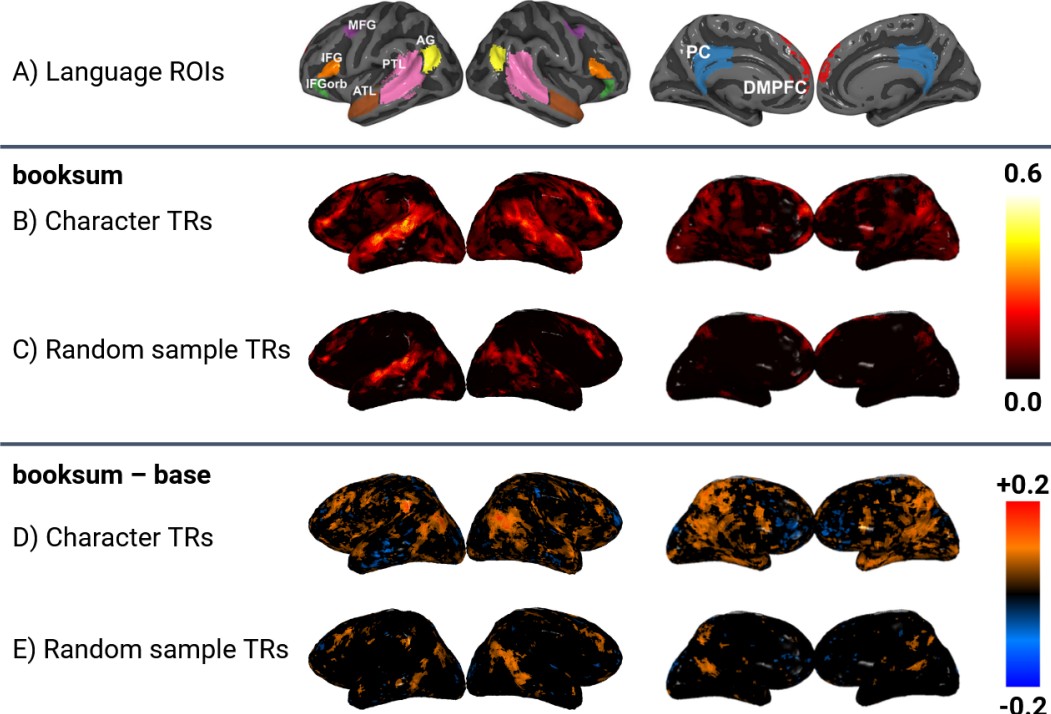

Figure 4: Pearson correlation averages between fMRI predictions from BigBird and true fMRI activity for one representative participant. Averages were computed over 8 layers for each model, and sequence lengths 20 to 500. (A) Visualization of regions of interest (ROIs) thought to underlie language comprehension. (B) Pearson correlations for booksum models for Character TRs. (C) Same as B, but for an equal number of TRs sampled randomly from all TRs. (D) Difference in Pearson correlations between booksum and base models, for Character TRs. (E) Same as D, but for the random sample. Comparing row B and C, there is greater brain alignment for Character TRs than for the random sample, both within and outside the language ROIs. Comparing row D and E, Character TRs improve in more voxels than the random sample TRs. Only voxels with Pearson correlation significantly greater than 0 are plotted (one-sample t-test, FDR corrected for multiple comparisons at significance level 0.05). Similar results for other participants are in Appendix K.

## 7 CONCLUSIONS AND FUTURE WORK

**Conclusions for brain-NLP interdisciplinary field.** How and why do brain and NLP representations align? This work contributes the following findings towards this question. (1) Understanding of characters and other discourse features is a significant factor in brain-NLP alignment. (2) Deeper understanding of a text contributes to brain-NLP alignment independently from language modeling (LM) ability. Overall, our work helps to fill a knowledge gap in the brain-NLP field.

**Conclusions for NLP.** Language modeling (LM) has achieved great success and popularity. However, is LM sufficient to develop NLP models with deep language understanding? We present a unique approach to answer this question by drawing insights from human brains. Our results reveal that: (1) LM models achieve poorer representations for Characters and other discourse features compared to deeper understanding models. (2) Existing training methods for narrative understanding are a first step towards developing language models with deep language understanding. Using results from brain-NLP alignment, we argue for the need to train NLP models with deeper understanding of long-range context, going beyond language modeling.

**Future work.** There are multiple exciting avenues to build on our work. (1) We showed that brain alignment differs across discourse features. Future work can investigate why these differences exist. (2) We showed that deeper understanding in NLP models improves brain alignment. Future work can explore the mechanisms behind deeper understanding of texts in NLP models.

## 8 ACKNOWLEDGMENTS

We wish to thank Gabriele Merlin for valuable comments and discussions.

## 9 REPRODUCIBILITY STATEMENT

Code available at: `https://github.com/awwkl/brain_language_summarization`. In this repository, we provide the following: (1) Code used to reproduce the results in our paper. (2) README explaining how to install packages, preprocess data, run experiments, and plot visualizations of results. (3) In our paper, we run a large number of experiments (many models, layers, sequence lengths, subjects, discourse features, brain ROIs, etc). Hence, we provide scripts to automate the process of running experiments across the various models, layers, etc. We hope this will make it easy for others to use our code efficiently. (4) After running experiments to obtain data, you can also use our code for plotting the figures in our paper, as well as additional graphs to visualize results. We hope this will allow others to use our code for future projects.

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

## A    NLP MODELS USED IN OUR WORK

For all 4 base models and 4 booksum models, we use pretrained models provided by Hugging Face (Wolf et al., 2020).

- BART (Lewis et al., 2019)
  - Base model: https://huggingface.co/facebook/bart-base
  - Booksum model: https://huggingface.co/KamilAin/bart-base-booksum
- LED (Beltagy et al., 2020)
  - Base model: https://huggingface.co/allenai/led-base-16384
  - Booksum model: https://huggingface.co/pszemraj/led-base-book-summary
- BigBird (Zaheer et al., 2020)
  - Base model: https://huggingface.co/google/bigbird-pegasus-large-bigpatent
  - Booksum model: https://huggingface.co/pszemraj/bigbird-pegasus-large-K-booksum
- LongT5 (Guo et al., 2021)
  - Base model: https://huggingface.co/google/long-t5-tglobal-base
  - Booksum model: https://huggingface.co/pszemraj/long-t5-tglobal-base-16384-book-summary

## B    EXAMPLES FROM BOOKSUM DATASET

Here, we describe the inputs and outputs used for training models on BookSum (Kryscinski et al., 2021). The input documents are chapters from books. The output summaries are human-written, obtained from various sources such as CliffNotes, SparkNotes, GradeSaver, etc. Below are examples of summaries from BookSum.

**Summary of Chapter 18 of "Sense and Sensibility" by Jane Austen.**
Edward's reticence became more noticeable as his visit continued. On one occasion Marianne attempted to leave the couple alone together and met with the speedy withdrawal of Edward from the room where she had left them. One day when the sisters and Edward were having breakfast, Marianne noticed "a ring, with a plait of hair in the centre," on one of Edward's fingers. She asked him if it was Fanny's, and Edward answered in the affirmative. Elinor and Marianne, however, believed it was Elinor's, although while Marianne thought it was freely given, Elinor was "conscious it must have been procured by some theft or contrivance unknown to herself." At noon, Sir John and Mrs. Jennings arrived, curious about the guest at the cottage. Realizing that he was the mysterious man with a name beginning with "F," Sir John insisted that they must "drink tea" at Barton Park that night and dine there the following day.

**Summary of Chapter 4 of "The Picture of Dorian Gray" by Oscar Wilde.**
Over the next several years, Dorian becomes obsessed with the book given to him by Lord Henry. He buys multiple copies of the "first edition, and them bound in different colors so that they might suit his moods." To Dorian, "the whole book...seemed to contain the story of his own life, written before he had lived it." Like the book's young hero, Dorian begins immersing himself in varied interests, including religion, mysticism, music, jewels, ancient tapestries, and the study of his own ancestors. Dorian is, however, quick to change obsessions once they no longer interest him, following the whims of his desire with the passion of an artist. He clings to each current obsession fervently, studying it and acquiring as many fanciful examples of it as he can find. He buys extravagent gowns covered in hundreds of pearls to feed his interest in jewels, and ancient, golden-threaded tapestries to nourish his curiosity about embroidery. As soon as a given subject has exhausted itself in his mind, however, he drops it in favor of his next interest. For the next 18 years, capriciousness is a way of life for Dorian...

**Summary of Chapter 6 of "The Brothers Karamazov, Volume 3" by Fyodor Dostoevsky.**
When Alyosha enters his father's home, his father and brother Ivan are finishing dinner and the servants attend them in the dining room. Here we're given a little background on Smerdyakov, Stinking Lizaveta's son. A quiet, sullen child, Smerdyakov was discovered to have the "falling sickness" ; he would fall into fits every month or so. For some reason Fyodor grew quite fond of the

boy after his illness was discovered. One day, Smerdyakov was found picking through his soup, and Fyodor decided that Smerdyakov should be trained as a chef. After training in Moscow, Smerdyakov came back to be Fyodor's cook. Now 24, he is just as sullen and silent as ever.

## C    FURTHER DETAILS ABOUT BRAIN ALIGNMENT APPROACH

In this section, we describe additional details for aligning the word-level NLP representations to human brain activity. We follow a general alignment approach that has been previously used in several works (Jain & Huth, 2018; Toneva & Wehbe, 2019; Schrimpf et al., 2021).

We start with the following variables. NLP representations $\in \mathbb{R}^{5176 \times d}$, where $d$ is the embedding size. Brain activity $\in \mathbb{R}^{n \times v_i}$, where $n$ is the number of TRs and $v_i$ is the number of voxels in the brain of participant $i$.

First, we reduce the dimensionality of the word-level NLP representations $\in \mathbb{R}^{5176 \times d}$ using PCA and retain the top 10 principle components (more than $75\%$ of the variance) to result in a matrix $\in \mathbb{R}^{5176 \times 10}$. Next, we construct TR-level NLP representations by down-sampling the word embeddings (words presented at 0.5 seconds) to the TR rate (2 seconds) by averaging to corresponding word-level NLP representations to result in a matrix $\in \mathbb{R}^{n \times 10}$, where $n$ is the number of TRs. The final NLP representations are constructed by concatenating the TR-level NLP representations corresponding to the previous 4 TRs. The previous TR-level embeddings are included to account for the lag in the hemodynamic response that fMRI records. Because the response measured by fMRI is an indirect consequence of brain activity that peaks about 6 seconds after stimulus onset, predictive methods commonly include preceding time points (Nishimoto et al., 2011; Wehbe et al., 2014a; Huth et al., 2016). This allows for a data-driven estimation of the hemodynamic response functions (HRFs) for each voxel, which is preferable to assuming one because different voxels may exhibit different HRFs. These steps produce the final NLP inputs to the brain alignment stage $X_{l,s}^m \in \mathbb{R}^{n \times 40}$, for model $m$, layer $l$, sequence length $s$.

After reducing the dimensionality of NLP representations, we learn a function that uses NLP representations $X_{l,s}^m \in \mathbb{R}^{n \times 40}$ to predict brain activity $\in \mathbb{R}^{n \times v_i}$. Similarly to previous work, we parameterize this function as a linear function, regularized using the ridge penalty. We train the function in a cross-validated way and test its performance on the data that was heldout during training. We select the ridge parameter via nested cross-validation. Since the fMRI data was collected in 4 runs of approximately equal length, we use 4-fold cross-validation where each fold corresponds to holding out one run of fMRI data for testing. We also remove 10TRs of fMRI data and the corresponding NLP representations from the ends of each test run to avoid possible train-test overlap.

Finally, we evaluate the alignment between the NLP predictions and the true fMRI activity, as described in the main paper.

## D    INTERPRETABILITY APPROACH TO COMPARE BRAIN ALIGNMENT ACROSS VARIOUS DISCOURSE FEATURES

---
**Algorithm 2** Interpretability approach to compare brain alignment across discourse features

---
1:  **Input:** NLP encoding predictions, fMRI activity          ▷ Both $\in \mathbb{R}^{n\ TRs \times v_i\ voxels}$
2:  **for all** discourse features **do**
3:      label words with discourse feature          ▷ One-hot vector $\in \mathbb{R}^{5176\ words}$
4:      map labeled words to fMRI TRs labeled with discourse feature       ▷ One-hot vector $\in \mathbb{R}^n$
5:      extract NLP encoding predictions for TRs with the discourse feature      ▷ $\mathbb{R}^{160 \times v_i}$
6:      extract actual brain activity for TRs with the discourse feature      ▷ $\mathbb{R}^{160 \times v_i}$
7:      compute Pearson correlation between extracted NLP predictions and brain activity    ▷ $\mathbb{R}^{v_i}$
8:      **Result:** Pearson correlation score for discourse feature, for each voxel      ▷ $\mathbb{R}^{v_i}$
9:  **end for**
10: **Result:** Pearson score for all discourse features

---

We present the pseudocode for our interpretability approach in Algorithm 2, and elaborate here.

The Pearson correlation score is computed between a pair of NLP encoding predictions and actual brain activity. Both $\in \mathbb{R}^{n \times v_i}$, where $n$ is the number of TRs and $v_i$ is the number of voxels in the brain of participant $i$ (Line 1).

For each discourse feature (Line 2), we first label the words in the Harry Potter chapter that correspond to the discourse feature, producing a one-hot vector $\in \mathbb{R}^{5176}$. (Line 3) Then, we apply the mapping from the words to fMRI TRs, because fMRI values are recorded once for every few words. This produces a one-hot vector $\in \mathbb{R}^n$, which labels the TRs which contain the discourse feature (Line 4).

Different discourse features will have different number of TRs labeled, e.g., Characters has 236 TRs labeled, whereas Emotions has 170 TRs labeled. For each discourse feature, we sample 160 TRs out of the TRs labeled. We do this to ensure that we use the same number of TRs for each discourse feature, allowing a fair comparison of Pearson correlation scores between discourse features. Hence, we have a one-hot vector $\in \mathbb{R}^{160}$.

We extract the NLP encoding predictions and brain activity for only the TRs containing Characters, both $\mathbb{R}^{160 \times v_i}$ (Line 5 and 6). Finally, we compute their Pearson correlation, producing a vector $\mathbb{R}^{v_i}$. This produces one Pearson correlation score for each brain voxel (Line 7). To get the Pearson score for the Characters discourse feature, we take the average across all $v_i$ voxels.

Earlier, we mentioned that for each discourse feature, words in the Harry Potter chapter have been labeled. For example, the word "Draco" is marked with the Character feature, and the word "climbed" with the Motion feature. We used discourse feature labels from Wehbe et al. (2014a).

For each discourse feature, we computed the average Pearson correlation for booksum and base models. To compute these averages, we use all 8 models, 8 layers for each model, sequence lengths 20 to 1000, and all 8 human participants. The results are reported in 16.

## E   STATISTICAL TESTS AND FALSE DISCOVERY RATE (FDR) CORRECTION USING THE BENJAMINI–HOCHBERG (BH) PROCEDURE

We perform t-tests to identify which results are statistically significant (not simply due to chance). We used paired t-test to compare brain alignment between booksum and base models (e.g., Figure 1). We used one-sample t-test to identify brain voxels where a model's pearson correlation brain alignment is significantly greater than 0 (e.g., Figure 4).

As we perform a large number of statistical tests, we obtain a large number of p-values. Thus, some will have p-values below 0.05 purely by chance. To correct this problem of false discovery, we adjust the false discovery rate (FDR) using the Benjamini-Hochberg (BH) procedure. For each original p-value, this produces a new, adjusted p-value. Finally, we compare the adjusted p-value against a significance level of 0.05 to identify statistically significant results. For our paper, we report only statistically significant results after FDR-BH correction.

## F   SELECTED LAYERS, SEQUENCE LENGTHS AND PARTICIPANTS

To compute 20v20 and Pearson averages, we could not run results for all layers and sequence lengths for all models, due to computational constraints.

**Layers.** For each model, we selected 8 representative layers evenly spread across all its layers. For BART, LED and LongT5, these models contained 12 layers (1 to 12), and we selected 8 layers (1, 2, 4, 6, 8, 10, 11, 12). For BigBird, it contains 32 layers (1 to 32), and we selected 8 layers (2, 6, 10, 14, 18, 24, 28, 32).

**Sequence lengths.** We selected representative sequence lengths from 1 to 1000, listed here: 1, 5, 20, 100, 200, 300, 400, 500, 700, 1000. For BART and LED, we used only sequence lengths up to 500, due to their shorter input token limits.

**Participants.** We used fMRI results from all 8 human participants.

## G   AGGREGATED 20V20 AND PEARSON CORRELATION RESULTS FOR ALL 4 PAIRS OF MODELS

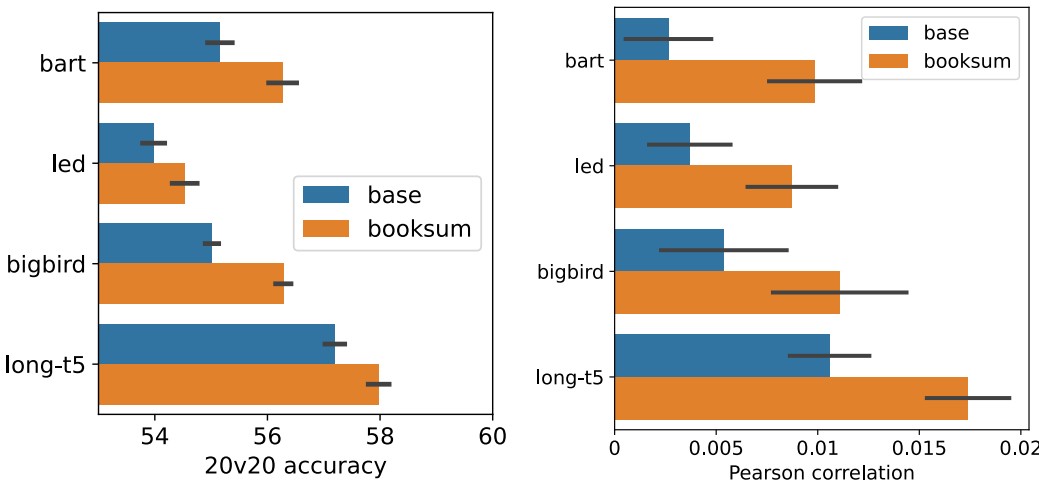

Figure 5: *Left*. Aggregated 20v20 brain alignment results for all 4 pairs of models. *Right*. Aggregated Pearson correlation brain alignment results for all 4 pairs of models. Averages were computed over 8 layers for each model, sequence lengths 1 to 1000, and all 8 participants. In both figures, we see that for each of the 4 pairs of models, the booksum model has significantly greater brain alignment than the base model (paired t-test, FDR corrected for multiple comparisons at significance level 0.05). These results show that training language models for deeper understanding improves brain alignment.

# H    RESULTS FOR 20v20 ACCURACY ACROSS SEQUENCE LENGTHS AND LAYERS

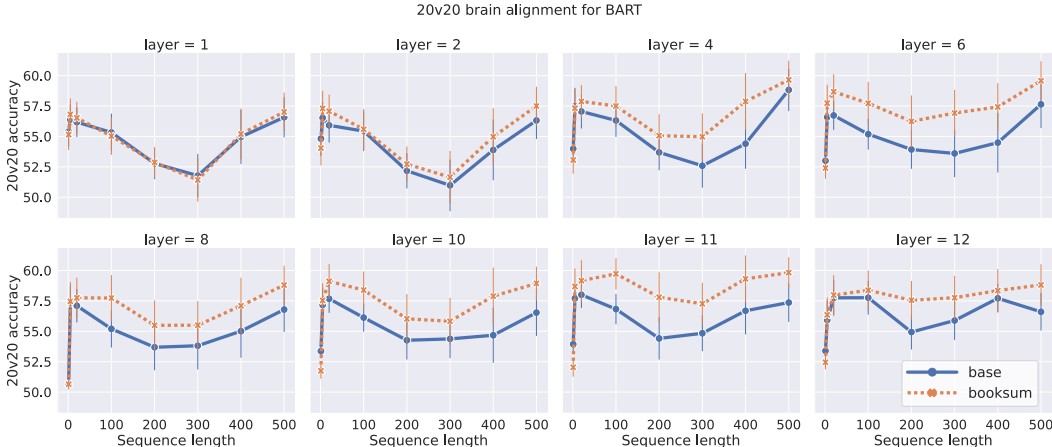

Figure 6: Comparing 20v20 classification accuracy between BART-base and BART-booksum. Averages were computed over all 8 human participants.

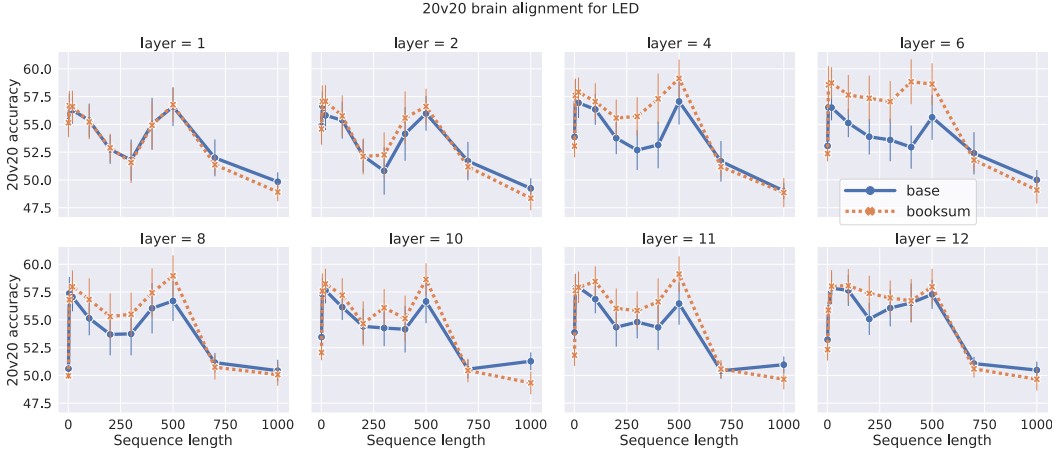

Figure 7: Comparing 20v20 classification accuracy between LED-base and LED-booksum. Averages were computed over all 8 human participants.

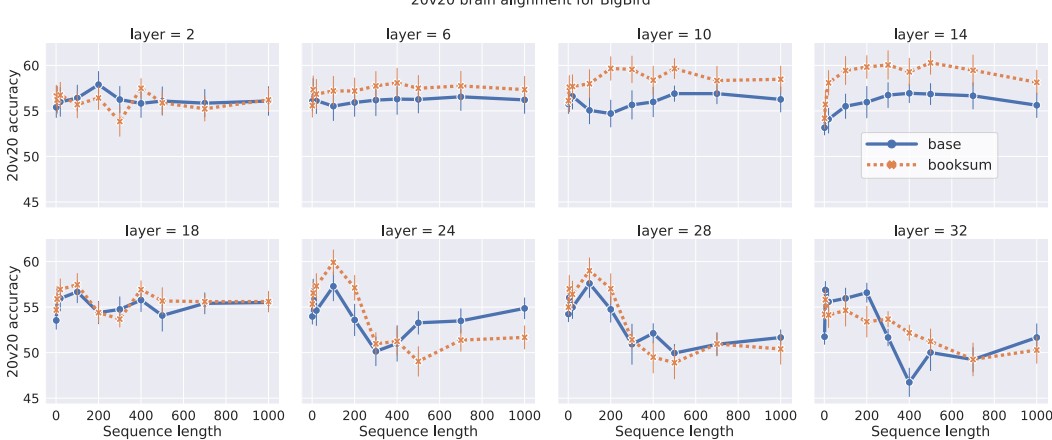

Figure 8: Comparing 20v20 classification accuracy between BigBird-base and BigBird-booksum. Averages were computed over all 8 human participants.

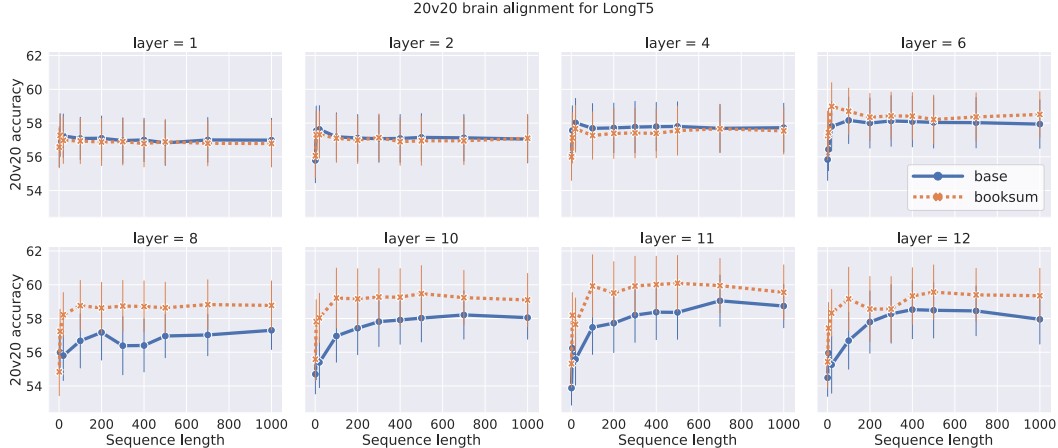

Figure 9: Comparing 20v20 classification accuracy between LongT5-base and LongT5-booksum. Averages were computed over all 8 human participants.

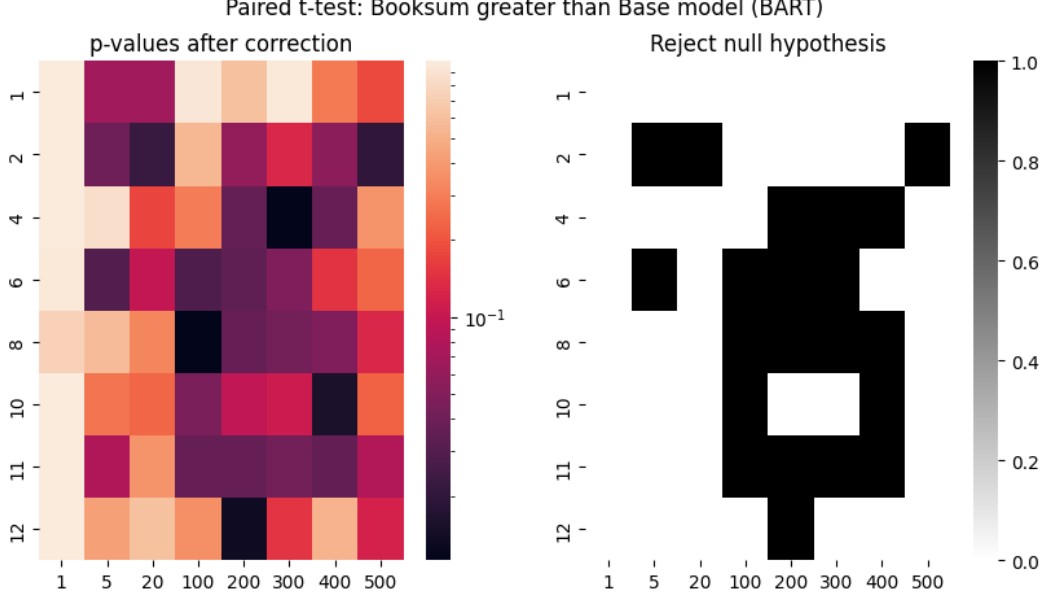

Figure 10: For BART, paired t-test was performed to identify the layers and sequence lengths where the booksum model has greater 20v20 brain alignment than the base model, with statistical significance. 20v20 averages were computed over all 8 human participants. *Left.* p-values obtained from paired t-test, after false discovery rate (FDR) correction using the Benjamini–Hochberg (BH) procedure. *Right.* At significance level 0.05, the layers and sequence lengths labeled in black are those where the booksum model has significantly greater brain alignment than the base model.

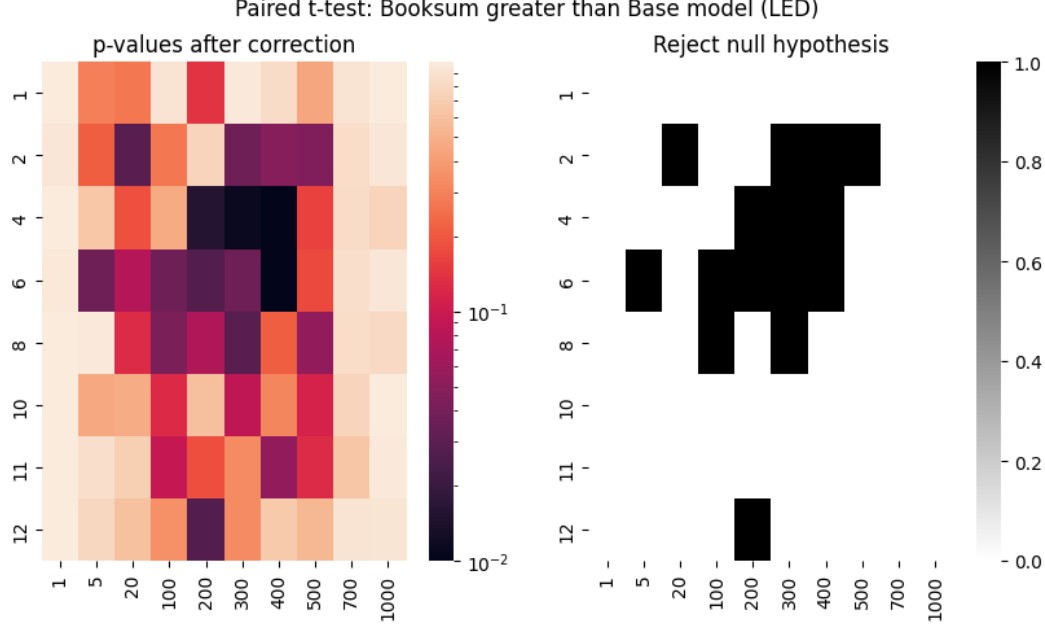

Figure 11: For LED, paired t-test was performed to identify the layers and sequence lengths where the booksum model has greater 20v20 brain alignment than the base model, with statistical significance. 20v20 averages were computed over all 8 human participants. *Left.* p-values obtained from paired t-test, after false discovery rate (FDR) correction using the Benjamini–Hochberg (BH) procedure. *Right.* At significance level 0.05, the layers and sequence lengths labeled in black are those where the booksum model has significantly greater brain alignment than the base model.

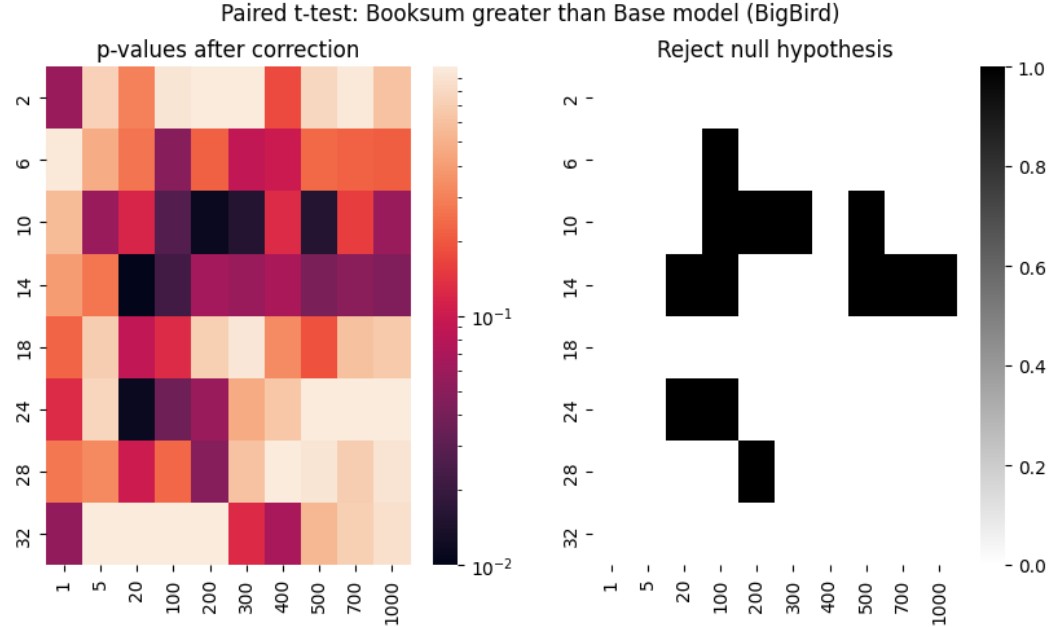

Figure 12: For BigBird, paired t-test was performed to identify the layers and sequence lengths where the booksum model has greater 20v20 brain alignment than the base model, with statistical significance. 20v20 averages were computed over all 8 human participants. *Left.* p-values obtained from paired t-test, after false discovery rate (FDR) correction using the Benjamini–Hochberg (BH) procedure. *Right.* At significance level 0.05, the layers and sequence lengths labeled in black are those where the booksum model has significantly greater brain alignment than the base model.

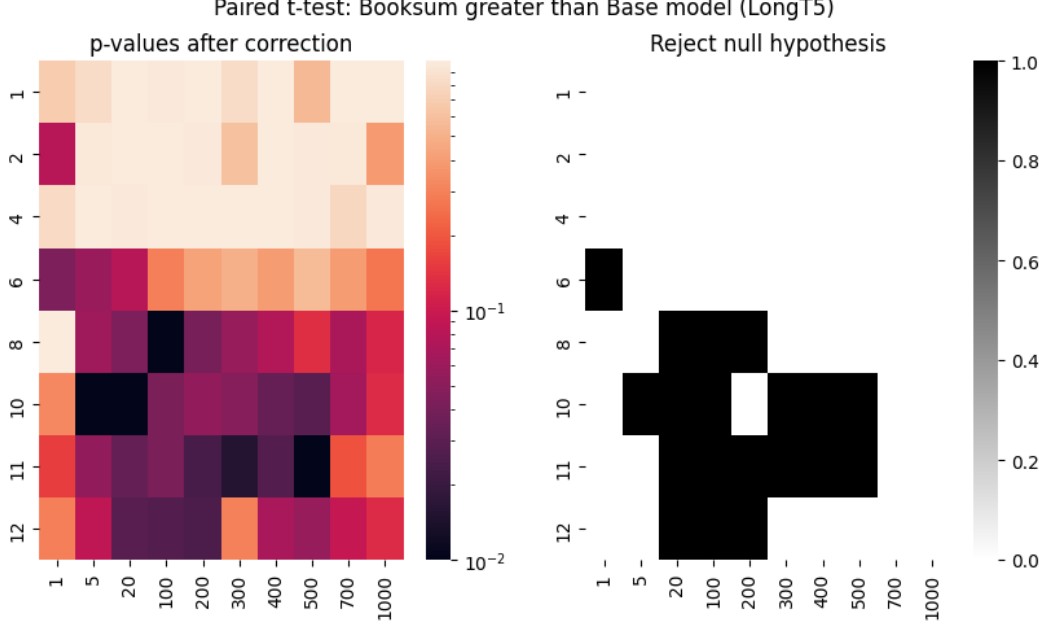

Figure 13: For LongT5, paired t-test was performed to identify the layers and sequence lengths where the booksum model has greater 20v20 brain alignment than the base model, with statistical significance. 20v20 averages were computed over all 8 human participants. *Left.* p-values obtained from paired t-test, after false discovery rate (FDR) correction using the Benjamini–Hochberg (BH) procedure. *Right.* At significance level 0.05, the layers and sequence lengths labeled in black are those where the booksum model has significantly greater brain alignment than the base model.

# I RESULTS FOR BRAIN REGIONS OF INTEREST (ROIS)

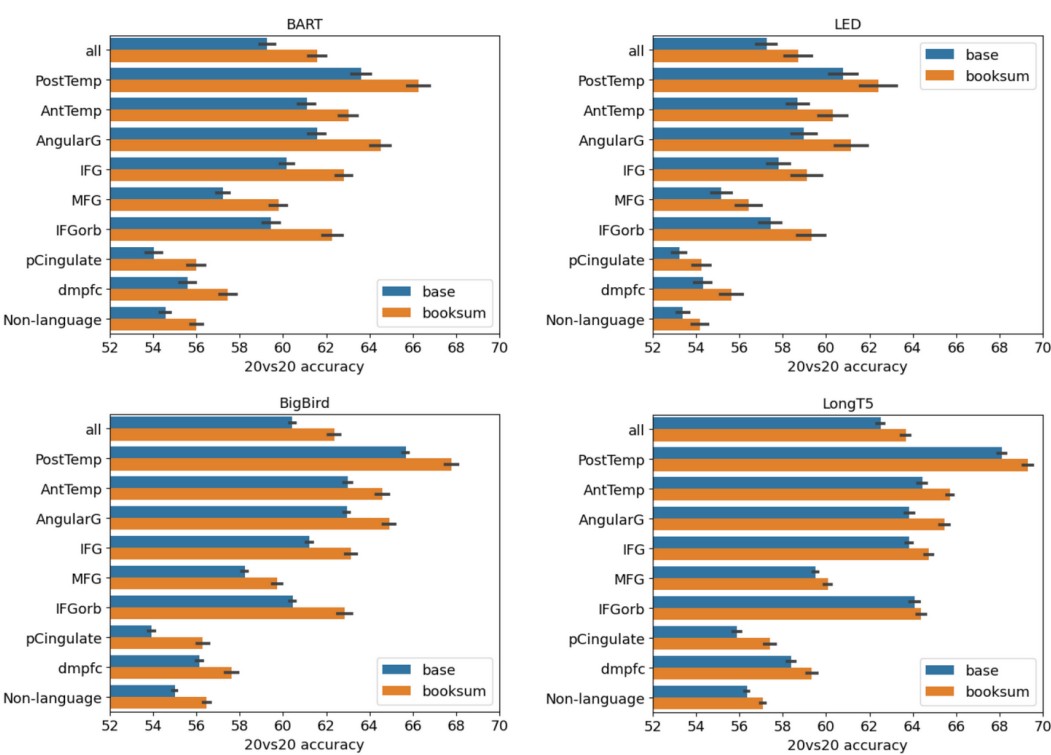

Figure 14: 20v20 accuracy averages for various brain Regions of Interest (ROIs), for each of the 4 models. Averages were computed over 8 layers for each model, sequence lengths 1 to 1000, and all 8 participants.

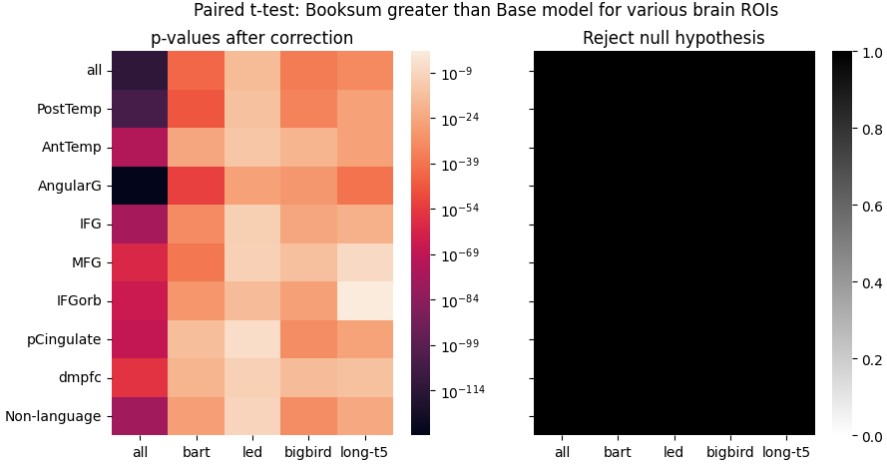

Figure 15: For each of the 4 models, paired t-test was performed to identify brain ROIs where the booksum model has greater 20v20 brain alignment than the base model, with statistical significance. We also perform the t-test for all the models combined together, labeled "all". 20v20 averages were computed over 8 layers for each model, sequence lengths 1 to 1000, and all 8 participants. *Left.* p-values obtained from paired t-test, after false discovery rate (FDR) correction using the Benjamini–Hochberg (BH) procedure. *Right.* At significance level 0.05, the brain ROIs labeled in black are those where the booksum model has significantly greater brain alignment than the base model.

## J  RESULTS FOR DISCOURSE FEATURES

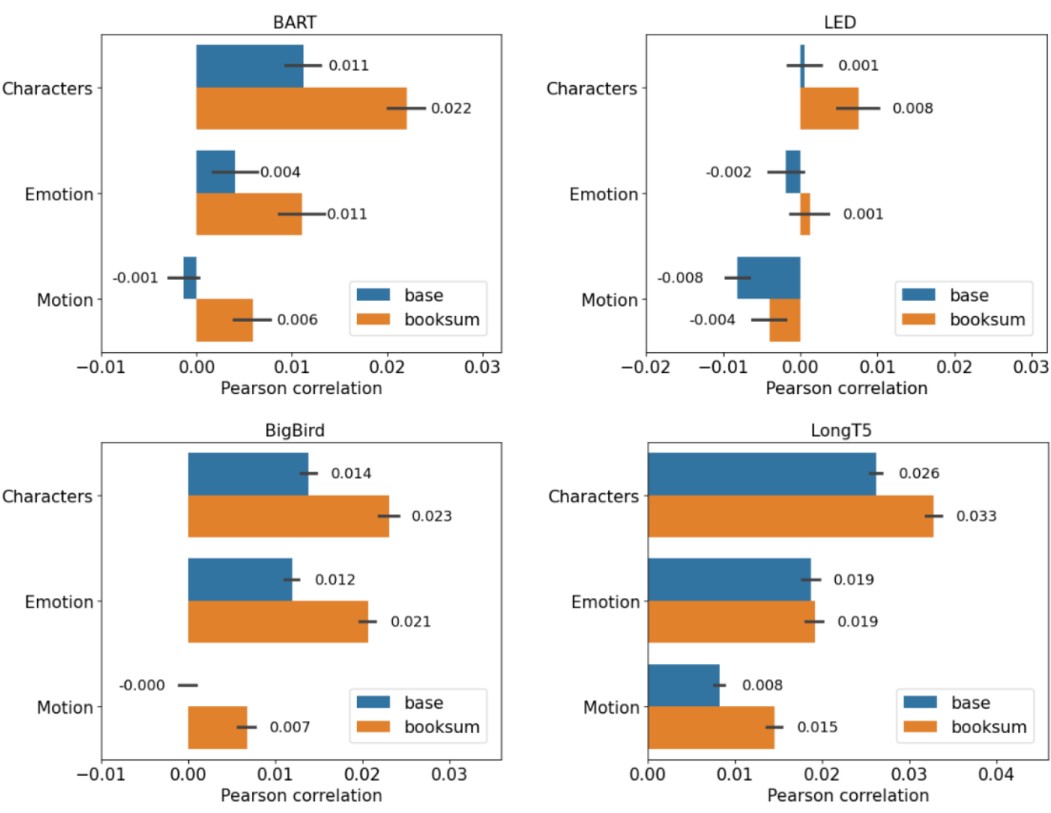

Figure 16: Pearson correlation brain alignment averages for various discourse features, for each of the 4 models. Averages were computed over 8 layers for each model, sequence lengths 20 to 500, and all 8 participants.

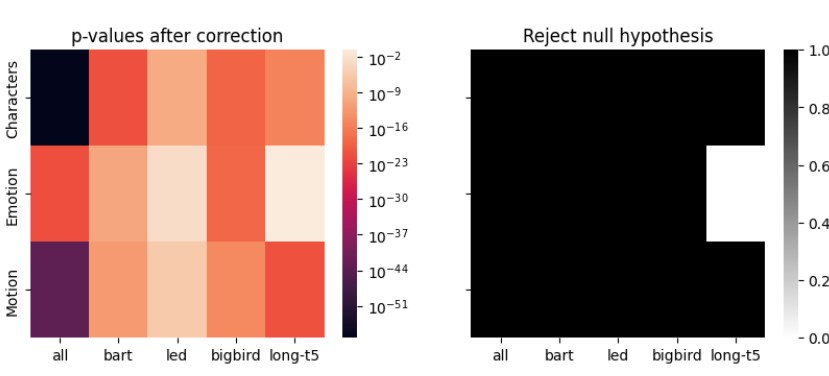

Figure 17: For each of the 4 models, paired t-test was performed to identify the discourse features where the booksum model has greater Pearson correlation brain alignment than the base model, with statistical significance. We also perform the t-test for all the models combined together, labeled "all". Pearson correlation averages were computed over 8 layers for each model, sequence lengths 20 to 500, and all 8 participants. *Left.* p-values obtained from paired t-test, after false discovery rate (FDR) correction using the Benjamini–Hochberg (BH) procedure. *Right.* At significance level 0.05, the discourse features labeled in black are those where the booksum model has significantly greater brain alignment than the base model.

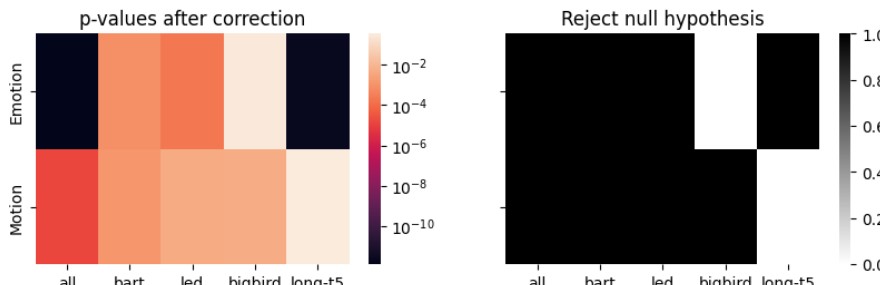

Figure 18: For each of the 4 models, paired t-test was performed to identify whether there were greater improvements in Pearson correlation brain alignment for Characters than other discourse features, with statistical significance. We also perform the t-test for all the models combined together, labeled "all". Pearson correlation averages were computed over 8 layers for each model, sequence lengths 20 to 500, and all 8 participants. *Left.* p-values obtained from paired t-test, after false discovery rate (FDR) correction using the Benjamini–Hochberg (BH) procedure. *Right.* At significance level 0.05, the brain ROIs labeled in black are those where the booksum model has significantly greater brain alignment than the base model.

# K RESULTS FOR BRAIN PLOTS

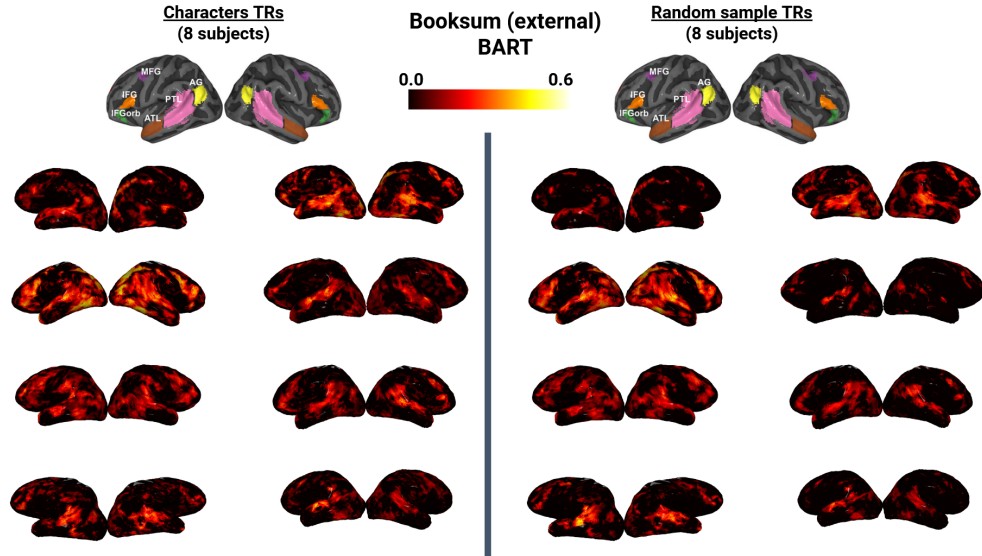

Figure 19: Pearson correlation averages of BART-booksum across brain voxels on the external brain surfaces of 8 human participants. Averages were computed over 8 layers for each model and sequence lengths 20 to 500. Only voxels with Pearson correlation significantly greater than 0 are plotted (one-sample t-test, FDR corrected for multiple comparisons at significance level 0.05). *Left.* Pearson correlation averages computed for only TRs which contain Characters. *Right.* Same as Left, but for an equal number of TRs sampled randomly from all TRs.

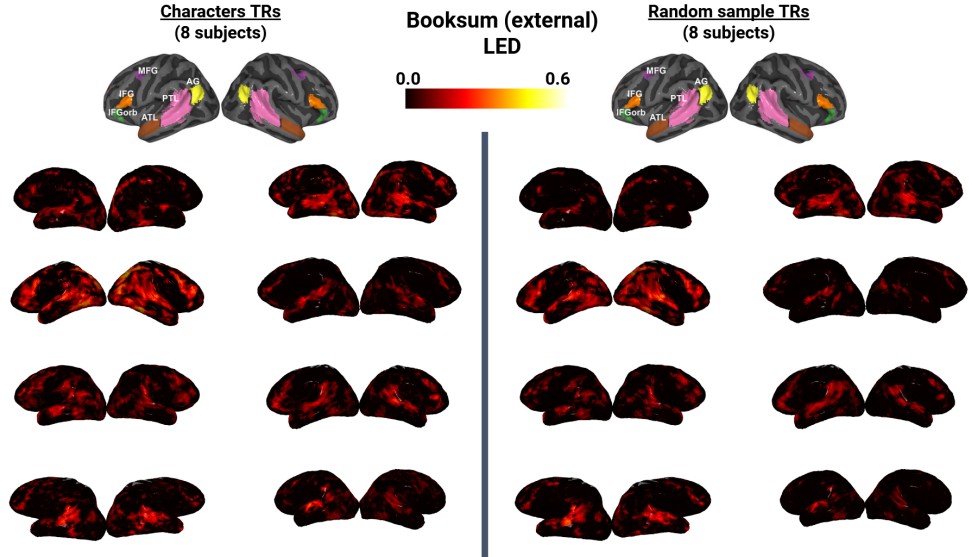

Figure 20: Pearson correlation averages of LED-booksum across brain voxels on the external brain surfaces of 8 human participants. Averages were computed over 8 layers for each model and sequence lengths 20 to 500. Only voxels with Pearson correlation significantly greater than 0 are plotted (one-sample t-test, FDR corrected for multiple comparisons at significance level 0.05). *Left.* Pearson correlation averages computed for only TRs which contain Characters. *Right.* Same as Left, but for an equal number of TRs sampled randomly from all TRs.

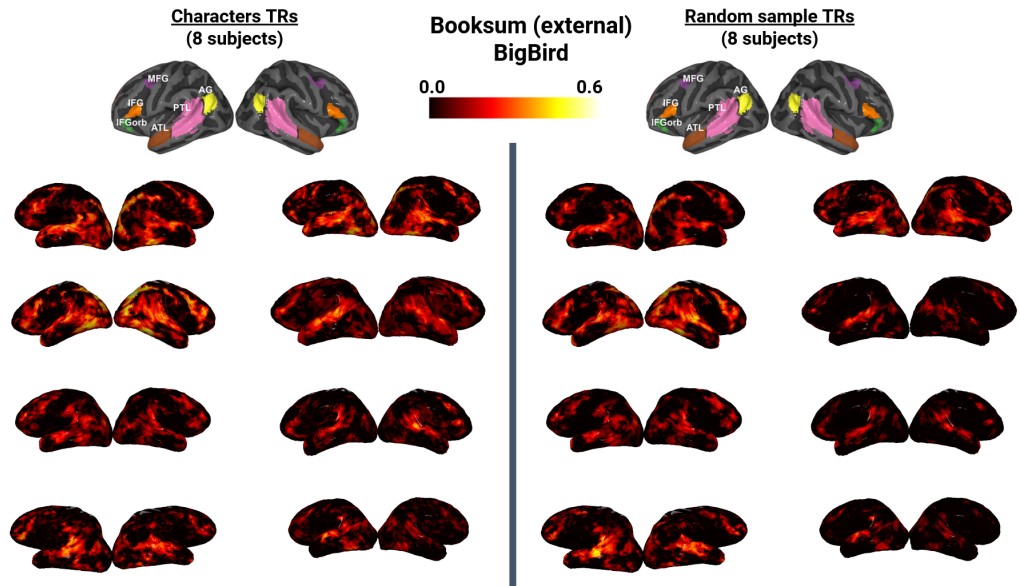

Figure 21: Pearson correlation averages of BigBird-booksum across brain voxels on the external brain surfaces of 8 human participants. Averages were computed over 8 layers for each model and sequence lengths 20 to 500. Only voxels with Pearson correlation significantly greater than 0 are plotted (one-sample t-test, FDR corrected for multiple comparisons at significance level 0.05). *Left.* Pearson correlation averages computed for only TRs which contain Characters. *Right.* Same as Left, but for an equal number of TRs sampled randomly from all TRs.

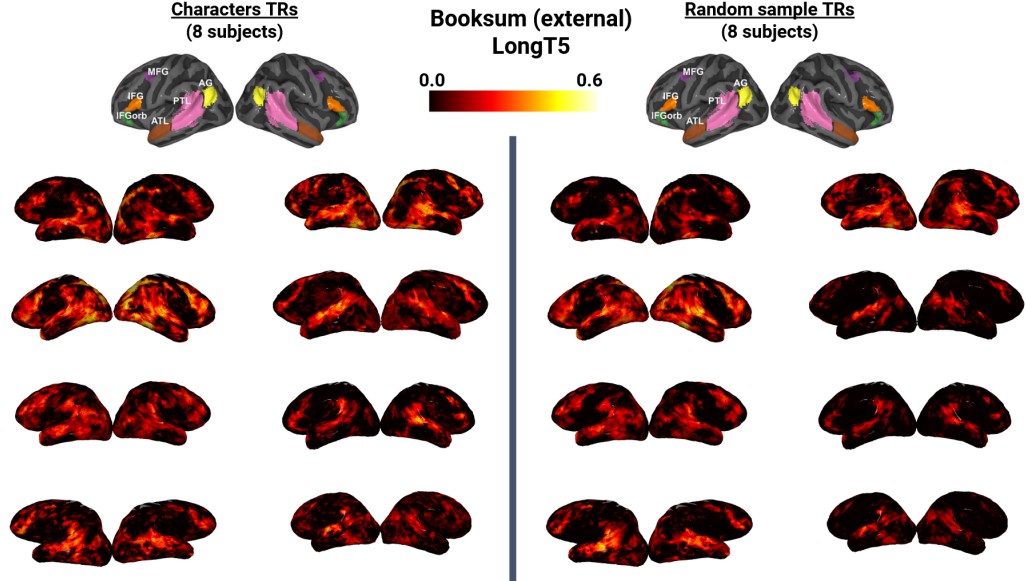

Figure 22: Pearson correlation averages of LongT5-booksum across brain voxels on the external brain surfaces of 8 human participants. Averages were computed over 8 layers for each model and sequence lengths 20 to 500. Only voxels with Pearson correlation significantly greater than 0 are plotted (one-sample t-test, FDR corrected for multiple comparisons at significance level 0.05). *Left.* Pearson correlation averages computed for only TRs which contain Characters. *Right.* Same as Left, but for an equal number of TRs sampled randomly from all TRs.

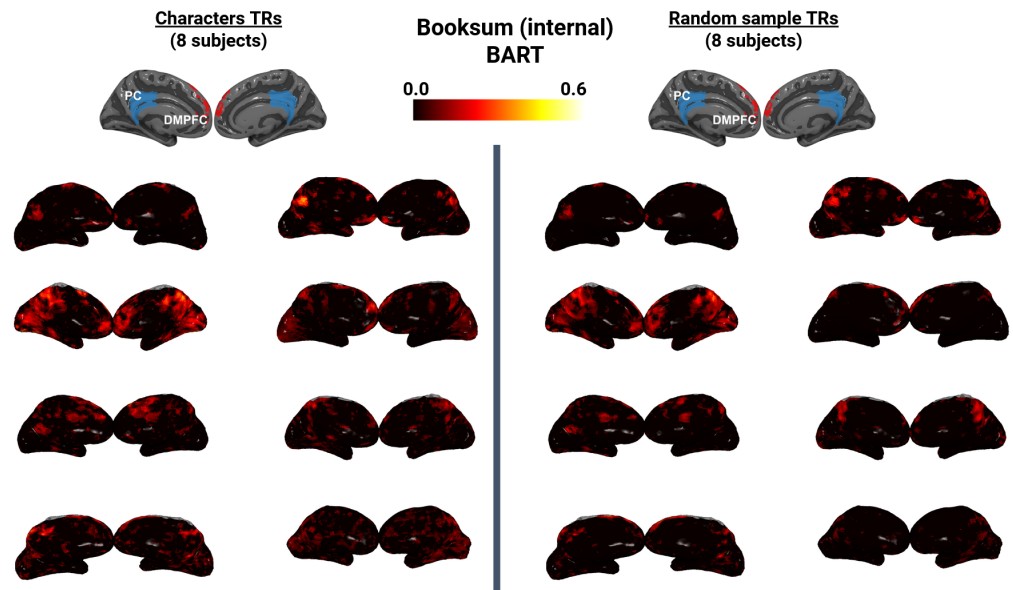

Figure 23: Pearson correlation averages of BART-booksum across brain voxels on the internal brain surfaces of 8 human participants. Averages were computed over 8 layers for each model and sequence lengths 20 to 500. Only voxels with Pearson correlation significantly greater than 0 are plotted (one-sample t-test, FDR corrected for multiple comparisons at significance level 0.05). *Left.* Pearson correlation averages computed for only TRs which contain Characters. *Right.* Same as Left, but for an equal number of TRs sampled randomly from all TRs.

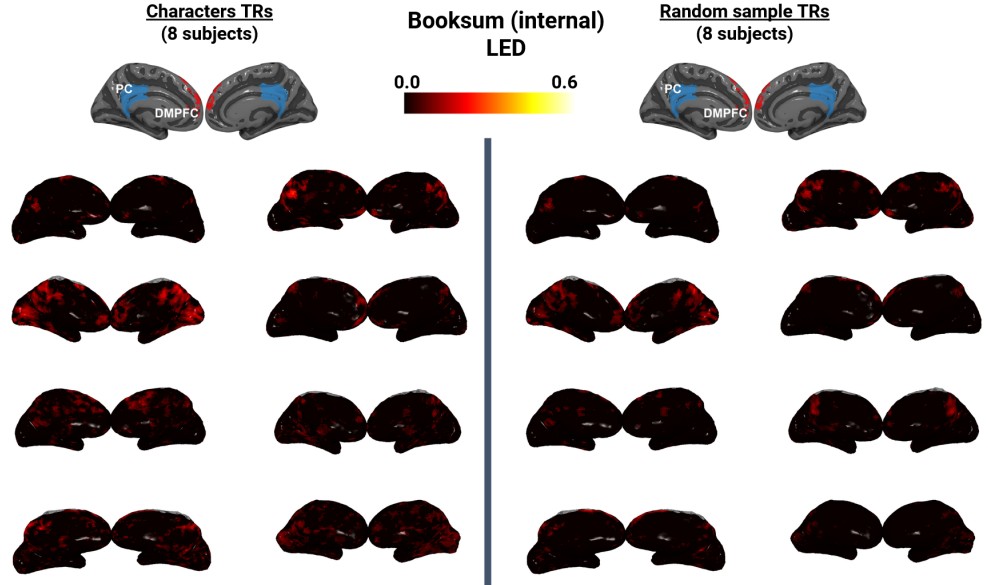

Figure 24: Pearson correlation averages of LED-booksum across brain voxels on the internal brain surfaces of 8 human participants. Averages were computed over 8 layers for each model and sequence lengths 20 to 500. Only voxels with Pearson correlation significantly greater than 0 are plotted (one-sample t-test, FDR corrected for multiple comparisons at significance level 0.05). *Left.* Pearson correlation averages computed for only TRs which contain Characters. *Right.* Same as Left, but for an equal number of TRs sampled randomly from all TRs.

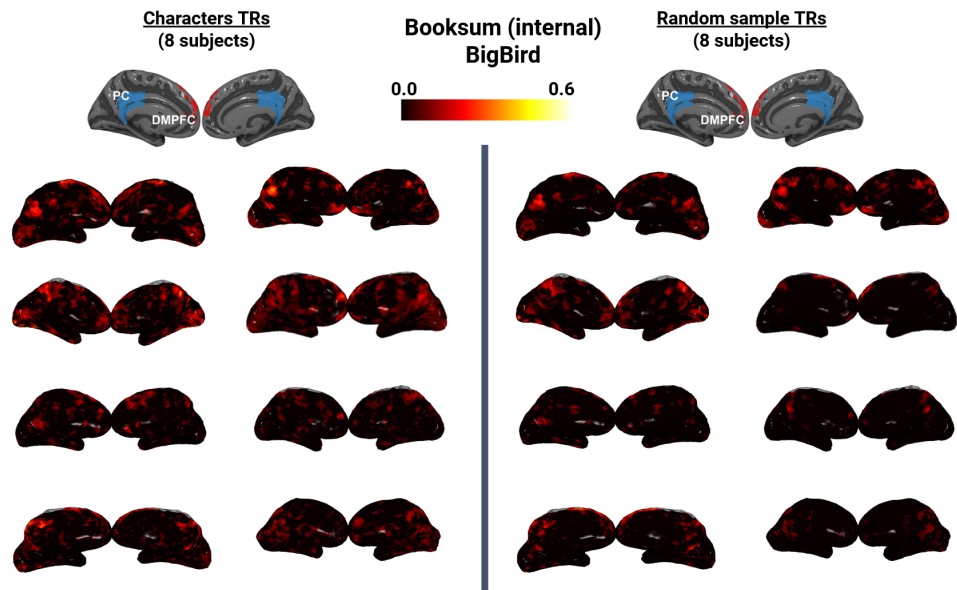

Figure 25: Pearson correlation averages of BigBird-booksum across brain voxels on the internal brain surfaces of 8 human participants. Averages were computed over 8 layers for each model and sequence lengths 20 to 500. Only voxels with Pearson correlation significantly greater than 0 are plotted (one-sample t-test, FDR corrected for multiple comparisons at significance level 0.05). *Left.* Pearson correlation averages computed for only TRs which contain Characters. *Right.* Same as Left, but for an equal number of TRs sampled randomly from all TRs.

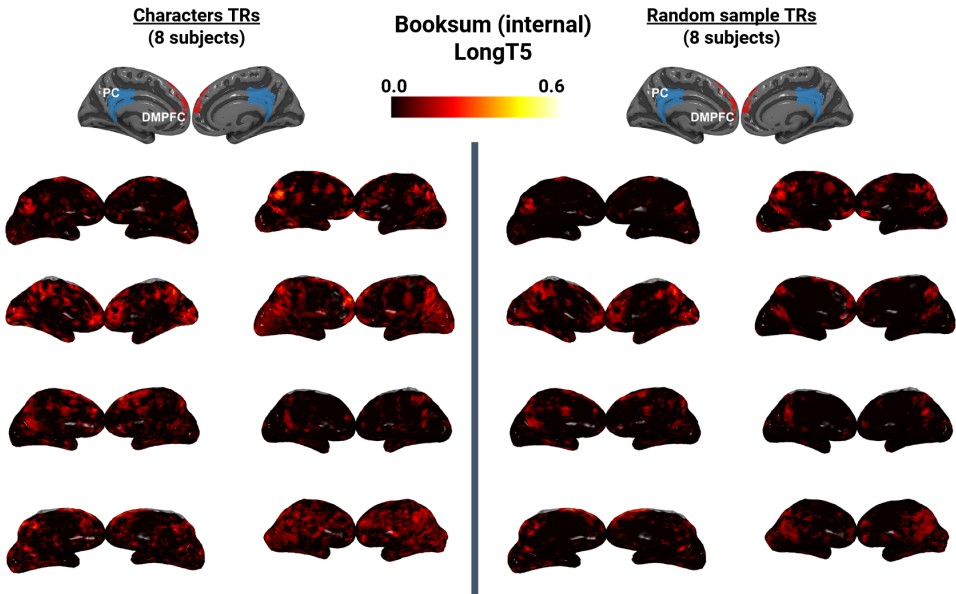

Figure 26: Pearson correlation averages of LongT5-booksum across brain voxels on the internal brain surfaces of 8 human participants. Averages were computed over 8 layers for each model and sequence lengths 20 to 500. Only voxels with Pearson correlation significantly greater than 0 are plotted (one-sample t-test, FDR corrected for multiple comparisons at significance level 0.05). *Left.* Pearson correlation averages computed for only TRs which contain Characters. *Right.* Same as Left, but for an equal number of TRs sampled randomly from all TRs.

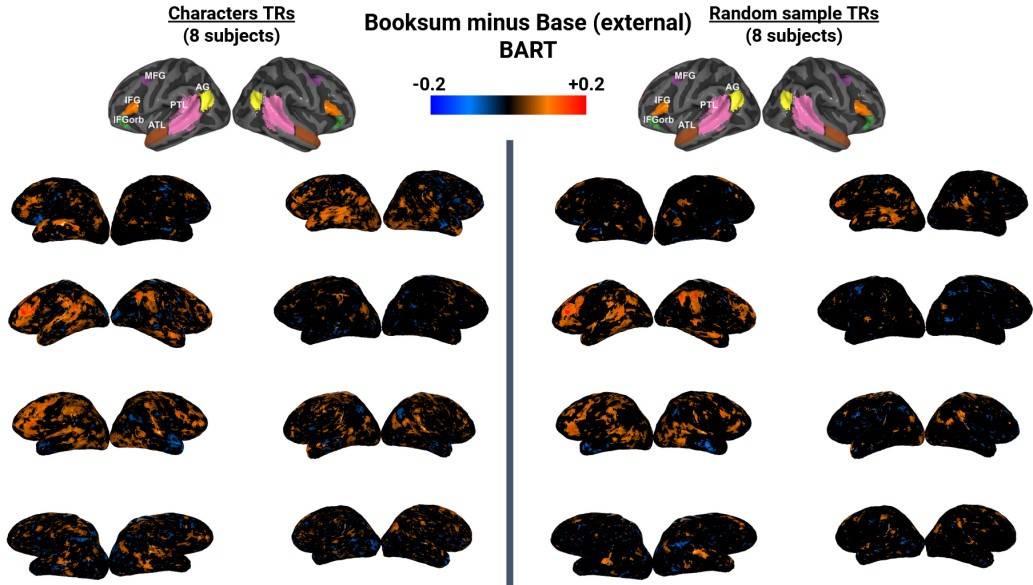

Figure 27: Difference in Pearson correlation averages between BART-booksum and BART-base across brain voxels on the external brain surfaces of 8 human participants. Averages were computed over 8 layers for each model and sequence lengths 20 to 500. Only voxels with significant Pearson correlation differences between the booksum and base model are plotted (paired t-test, FDR corrected for multiple comparisons at significance level 0.05). *Left.* Pearson correlation averages computed for only TRs which contain Characters. *Right.* Same as Left, but for an equal number of TRs sampled randomly from all TRs.

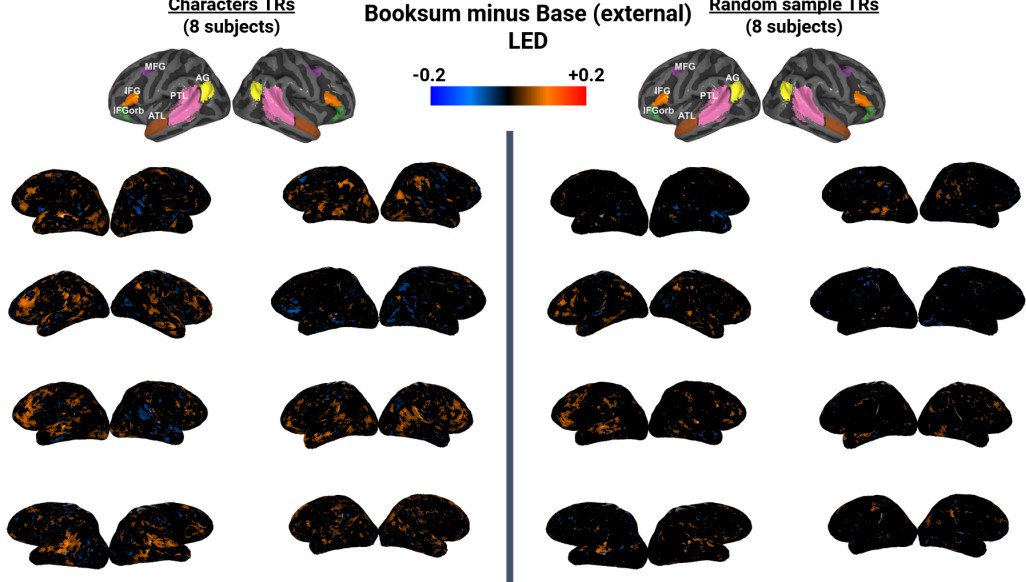

Figure 28: Difference in Pearson correlation averages between LED-booksum and LED-base across brain voxels on the external brain surfaces of 8 human participants. Averages were computed over 8 layers for each model and sequence lengths 20 to 500. Only voxels with significant Pearson correlation differences between the booksum and base model are plotted (paired t-test, FDR corrected for multiple comparisons at significance level 0.05). *Left.* Pearson correlation averages computed for only TRs which contain Characters. *Right.* Same as Left, but for an equal number of TRs sampled randomly from all TRs.

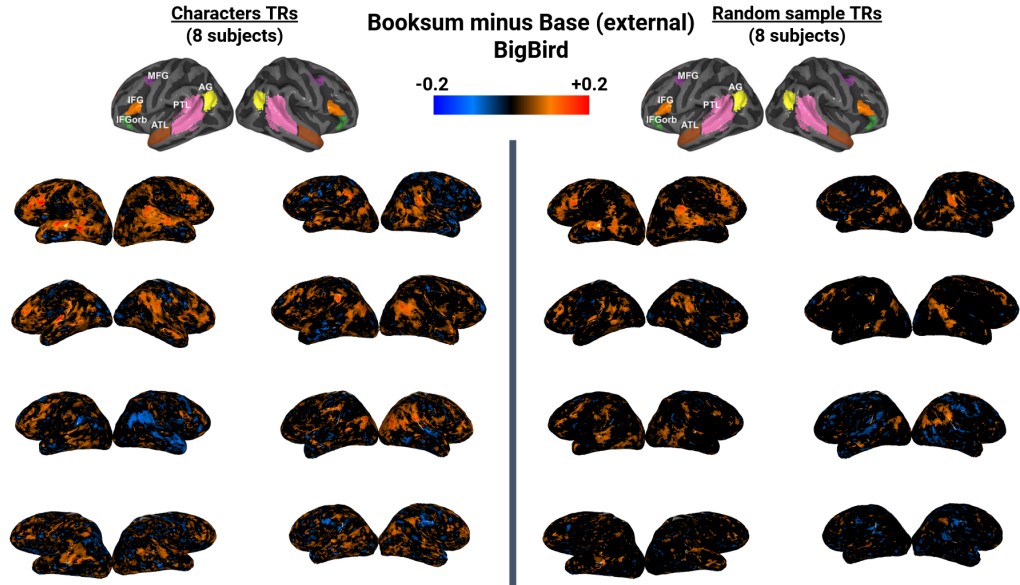

Figure 29: Difference in Pearson correlation averages between BigBird-booksum and BigBird-base across brain voxels on the external brain surfaces of 8 human participants. Averages were computed over 8 layers for each model and sequence lengths 20 to 500. Only voxels with significant Pearson correlation differences between the booksum and base model are plotted (paired t-test, FDR corrected for multiple comparisons at significance level 0.05). *Left.* Pearson correlation averages computed for only TRs which contain Characters. *Right.* Same as Left, but for an equal number of TRs sampled randomly from all TRs.

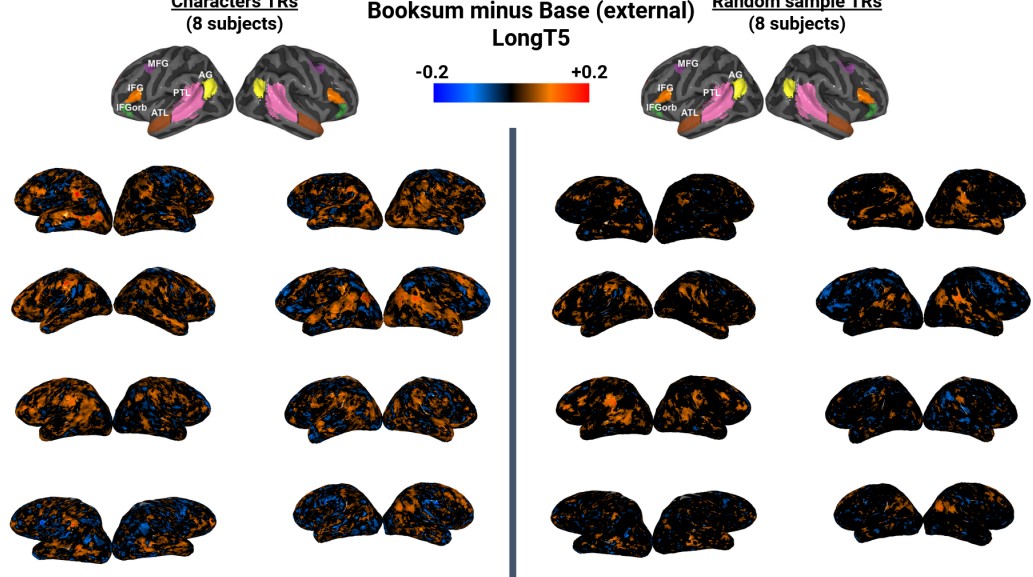

Figure 30: Difference in Pearson correlation averages between LongT5-booksum and LongT5-base across brain voxels on the external brain surfaces of 8 human participants. Averages were computed over 8 layers for each model and sequence lengths 20 to 500. Only voxels with significant Pearson correlation differences between the booksum and base model are plotted (paired t-test, FDR corrected for multiple comparisons at significance level 0.05). *Left.* Pearson correlation averages computed for only TRs which contain Characters. *Right.* Same as Left, but for an equal number of TRs sampled randomly from all TRs.

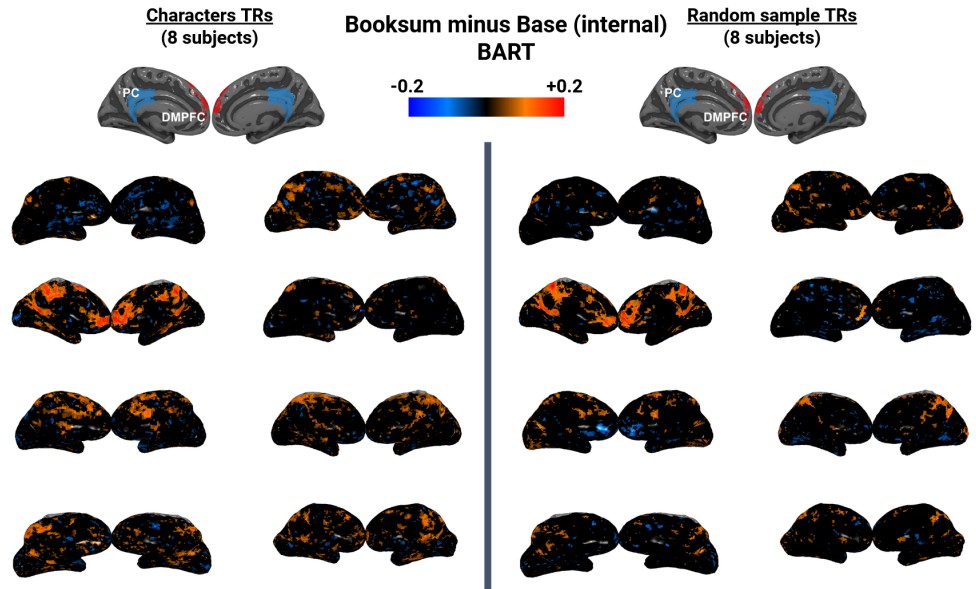

Figure 31: Difference in Pearson correlation averages between BART-booksum and BART-base across brain voxels on the internal brain surfaces of 8 human participants. Averages were computed over 8 layers for each model and sequence lengths 20 to 500. Only voxels with significant Pearson correlation differences between the booksum and base model are plotted (paired t-test, FDR corrected for multiple comparisons at significance level 0.05). *Left.* Pearson correlation averages computed for only TRs which contain Characters. *Right.* Same as Left, but for an equal number of TRs sampled randomly from all TRs.

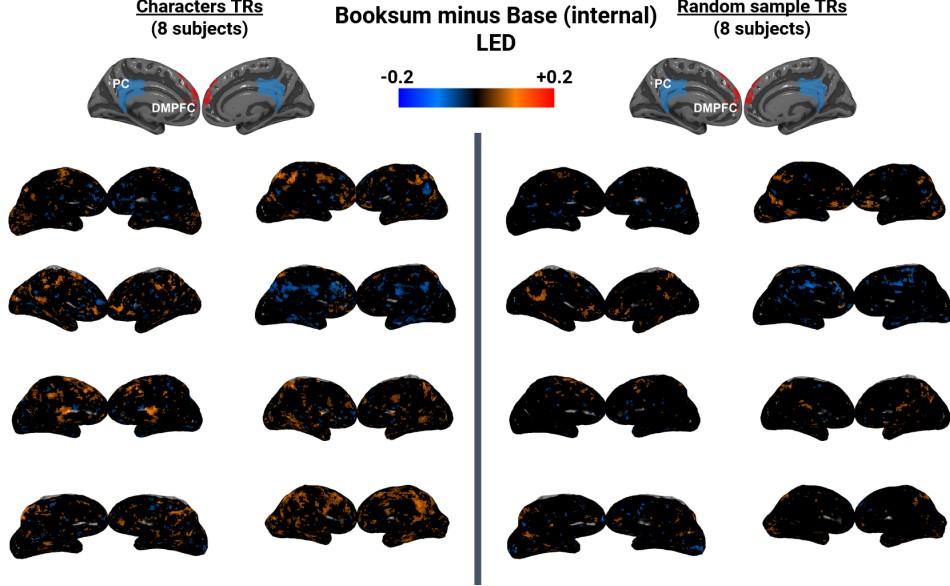

Figure 32: Difference in Pearson correlation averages between LED-booksum and LED-base across brain voxels on the internal brain surfaces of 8 human participants. Averages were computed over 8 layers for each model and sequence lengths 20 to 500. Only voxels with significant Pearson correlation differences between the booksum and base model are plotted (paired t-test, FDR corrected for multiple comparisons at significance level 0.05). *Left.* Pearson correlation averages computed for only TRs which contain Characters. *Right.* Same as Left, but for an equal number of TRs sampled randomly from all TRs.

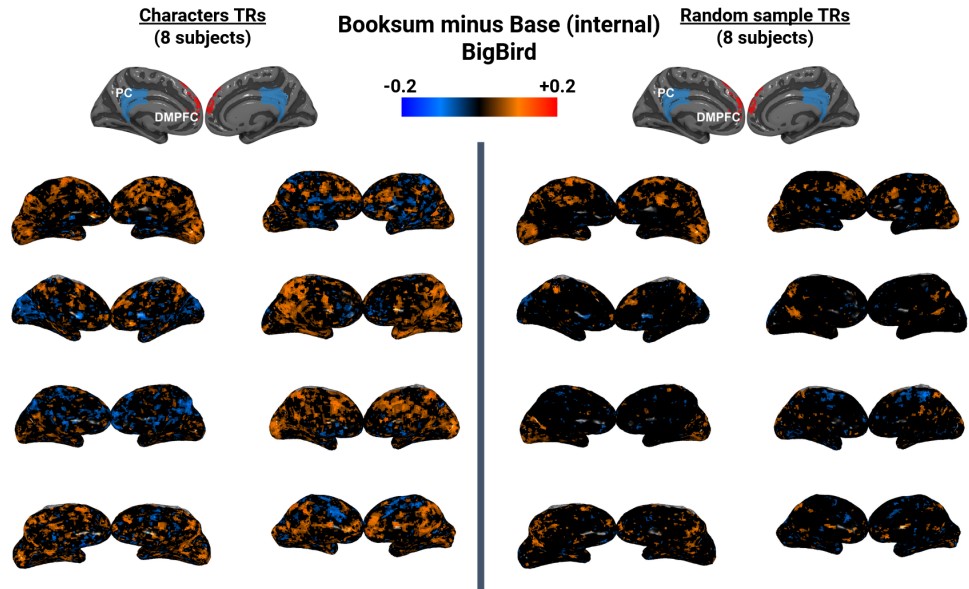

Figure 33: Difference in Pearson correlation averages between BigBird-booksum and BigBird-base across brain voxels on the internal brain surfaces of 8 human participants. Averages were computed over 8 layers for each model and sequence lengths 20 to 500. Only voxels with significant Pearson correlation differences between the booksum and base model are plotted (paired t-test, FDR corrected for multiple comparisons at significance level 0.05). *Left.* Pearson correlation averages computed for only TRs which contain Characters. *Right.* Same as Left, but for an equal number of TRs sampled randomly from all TRs.

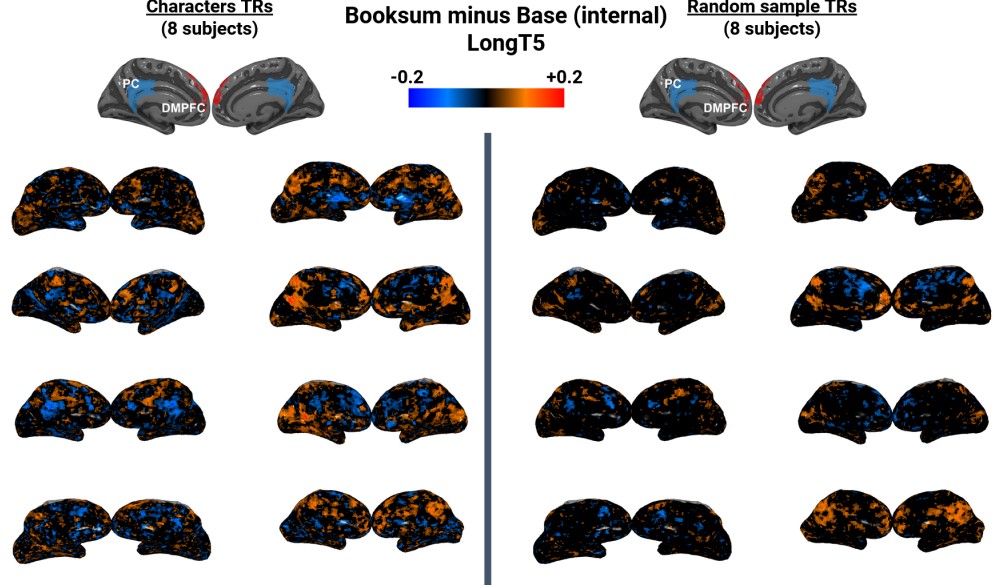

Figure 34: Difference in Pearson correlation averages between LongT5-booksum and LongT5-base across brain voxels on the internal brain surfaces of 8 human participants. Averages were computed over 8 layers for each model and sequence lengths 20 to 500. Only voxels with significant Pearson correlation differences between the booksum and base model are plotted (paired t-test, FDR corrected for multiple comparisons at significance level 0.05). *Left.* Pearson correlation averages computed for only TRs which contain Characters. *Right.* Same as Left, but for an equal number of TRs sampled randomly from all TRs.

## L    RESULTS FOR SUMMARIZATION MODELS ON NON-NARRATIVE DATASETS

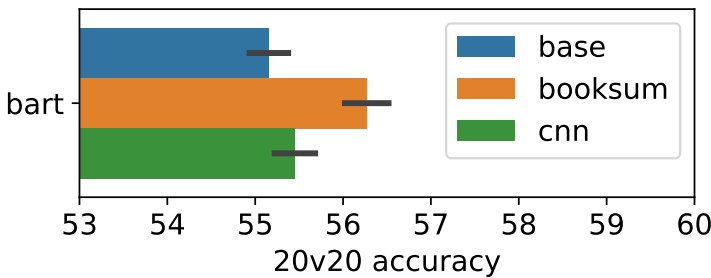

Figure 35: 20v20 averages for BART models. Averages were computed over 8 layers for each model, sequence lengths 1 to 500, and all 8 participants. The model trained to summarize long narratives (BART-booksum) has significantly greater brain alignment than the model pretrained with a language modeling objective (BART-base), as well as the model trained to summarize news articles (BART-cnn). Furthermore, BART-cnn only performs on par with the pretrained BART-base at brain alignment. This shows that improved brain alignment in BART-booksum over BART-base is not simply due to the summarization objective. We used paired t-tests, with FDR corrected for multiple comparisons at significance level 0.05.

Our paper shows improved brain alignment for booksum models over base models. There are 3 possible explanations: (1) similarity of narrative domain in train and test set, (2) summarization learning objective, (3) summarization on narrative domain, resulting in deeper understanding of characters, emotions and other discourse features. We argue that the third is the result: a deeper understanding emerges when training these models to summarize narrative texts.

In Appendix M, we show that improved brain alignment is not simply due to the first possibility: domain similarity. In this appendix, we show that improved brain alignment is not simply due to the second possibility: summarization learning objective.

We compared the brain alignment for 3 models: BART-base, BART-booksum and BART-cnn. BART-base was pretrained by corrupting documents and learning to reconstruct the original text, on a combination of books (narratives) and Wikipedia data (see Appendix A). BART-booksum initializes weights of BART-base and was further trained to summarize long narratives on the BookSum summarization dataset. BART-cnn initializes weights of BART-base and was further trained to summarize news articles from the CNN Dailymail dataset.

The model trained to summarize long narratives (BART-booksum) has significantly greater brain alignment than the model pretrained with a language modeling objective (BART-base), as well as the model trained to summarize news articles (BART-cnn). Furthermore, BART-cnn only performs on par with the pretrained BART-base at brain alignment. This shows that improved brain alignment in BART-booksum over BART-base is not simply due to the summarization objective.

Both the input documents and output summaries in the BookSum dataset are substantially longer than those in the CNN Dailymail dataset, which promotes deeper long-range understanding. For the BookSum dataset, the input documents have an average of more than 5000 words, and output summaries have roughly 500 words on average. For the CNN Dailymail dataset, the input documents have 766 words on average while the summaries have 53 words on average.

The texts from the BookSum dataset contain richer discourse features. For example, characters develop over the course of the entire book through their thoughts, actions, and interactions with other characters. We show in Section 6 that the booksum models are developing richer representations of these discourse features.

## M   RESULTS FOR TRAINING ON BOOKSUM DATASET WITH A LANGUAGE MODELING OBJECTIVE

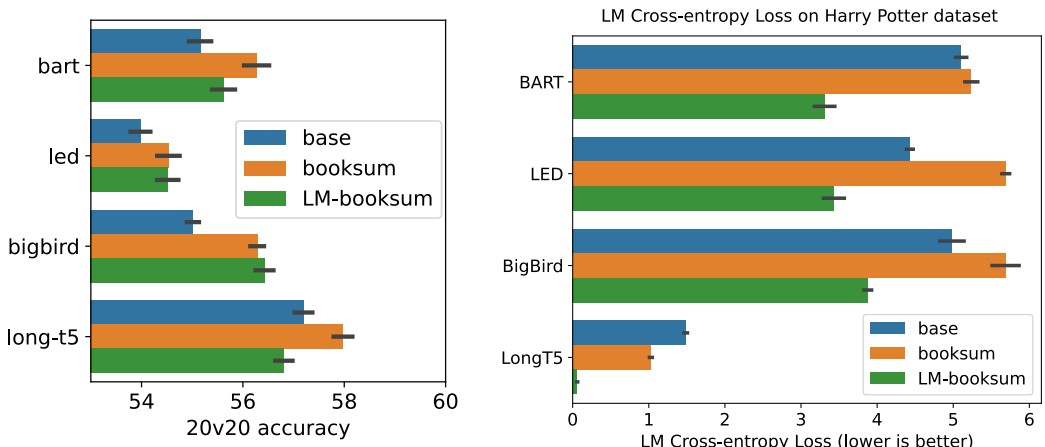

Figure 36: *Left.* 20v20 brain alignment for base, booksum and LM-booksum models. LM-booksum models were trained on the BookSum dataset with a language modeling objective. Averages were computed over 8 layers for each model, sequence lengths 1 to 1000, and all 8 participants. *Right.* Language modeling cross-entropy loss of all 4*3=12 models on the Harry Potter fMRI dataset.

For 2 models (BART, LongT5), the booksum variant has substantially better brain alignment than the LM-booksum counterpart. For the other 2 models (LED, BigBird), there is similar brain alignment for the booksum and LM-booksum variants. We used paired t-tests, with FDR corrected for multiple comparisons at significance level 0.05.

However, the LM-booksum variants have substantially better LM performance on the fMRI dataset than the booksum models. As LM performance has been shown to contribute to brain alignment (Schrimpf et al., 2021; Caucheteux & King, 2022; Goldstein et al., 2022), it is difficult to disentangle the brain alignment contributions of "deeper understanding" and LM performance in these models. The LM-booksum models have greater LM performance than both the base and booksum models, whereas the base and booksum models have similar LM performance. Hence, we believe that the LM-booksum models may not be a better baseline than base models.

Combined, these results suggest that the improved brain alignment is not simply due to domain similarity.

# N  RESULTS FOR GPT-2 - BOTH 20V20 AND PEARSON CORRELATION

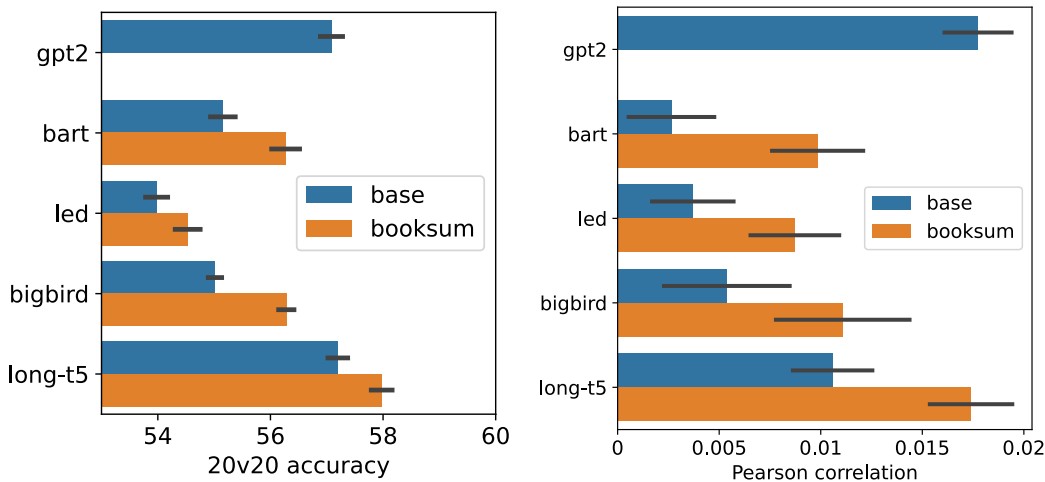

Figure 37: *Left.* Comparing average 20v20 accuracy for GPT-2 against the four pairs of base and booksum models. *Right.* Comparing average Pearson correlation for GPT-2 against the four pairs of base and booksum models. Averages were computed over 8 layers for each model, sequence lengths 1 to 1000, and all 8 participants.

We ran experiments to compute the aggregated 20v20 and Pearson correlation for GPT-2. For 20v20, LongT5-booksum has greater brain alignment than all other models, including GPT-2. For Pearson correlation, LongT5-booksum and GPT-2 have similar performance, and are both greater than the other models. We used paired t-tests, with FDR corrected for multiple comparisons at significance level 0.05.

Our work is focused on investigating how the training objective of summarizing narrative texts leads to improved brain alignment, hence we compared 4 pairs of base and booksum models. Attaining a model with SOTA brain alignment is not the goal of our work, hence we did not finetune GPT-2 on the booksum dataset. That is outside the scope of our project, but it may be an interesting direction for future research to pursue.

## O    RESULTS FOR TRAINING ON BALANCED DISCOURSE FEATURES

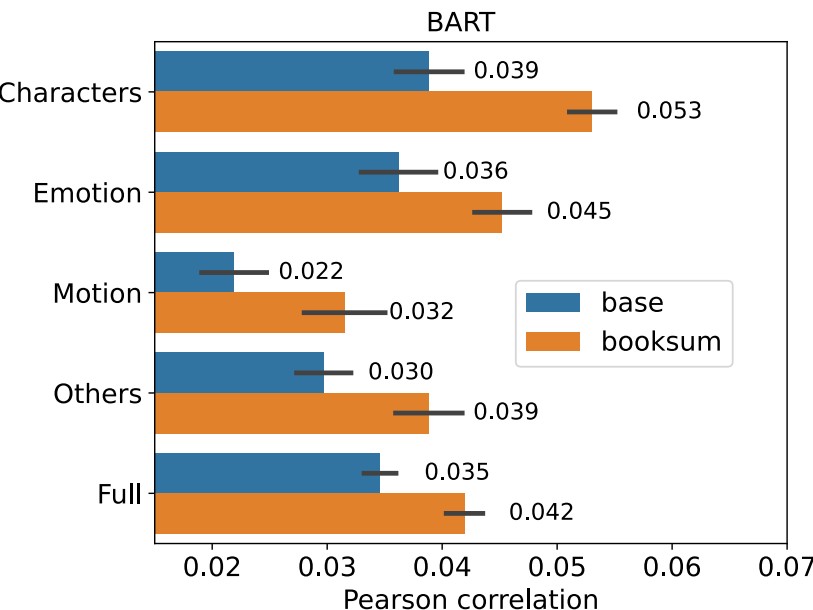

Figure 38: Pearson correlation averages for fMRI TRs corresponding to various discourse features. Averages were computed over 8 layers for each model, sequence lengths 20 to 500, and all 8 participants. "Others" refers to TRs which were not labeled with either "Characters", "Emotion" or "Motion". "Full" refers to the full set of all TRs. These plots were generated by training BART on the last 500 TRs of the Harry Potter fMRI dataset, where the distribution of discourse features was more balanced. BART-base and BART-booksum achieve greater brain alignment for Characters than other discourse features. When training BART for deep understanding, brain alignment significantly improves for all discourse features. However, it improves more for Characters than other discourse features. We used paired t-tests, with FDR corrected for multiple comparisons at significance level 0.05.

There are roughly 1200 fMRI TRs in the Harry Potter dataset where predictions are generated. Among this set, there are 242 Characters TRs, 174 Emotion TRs, 165 Motion TRs, and the remaining are TRs labeled as "Others". For each of the 3 discourse features, their TRs are relatively evenly distributed across the 1200 TRs. All 3 discourse features' TRs are also distributed in similar ways.

In Section 6, our results showed that Characters improved more (booksum minus base) than other discourse features when training models for deeper understanding. However, another possible cause of this result is that there are a different number of TRs for each discourse feature in the training set (242 for Characters, 174 for Emotion, 165 for Motion).

Hence, in this section, we ran additional experiments where we trained the linear encoding models on only the last 500 TRs, where the number of TRs for each discourse feature was more balanced (97 for Characters, 99 for Emotion, 72 for Motion). We then sampled an equal number of TRs (74) for each discourse feature during testing on the first 700 TRs. After controlling for an equal number of TRs in the training and test set across all discourse features, we still see the result that Characters improved more (booksum minus base) than other discourse features during training for deeper understanding.

## P    RESULTS FOR TRAINING LARGER MODELS

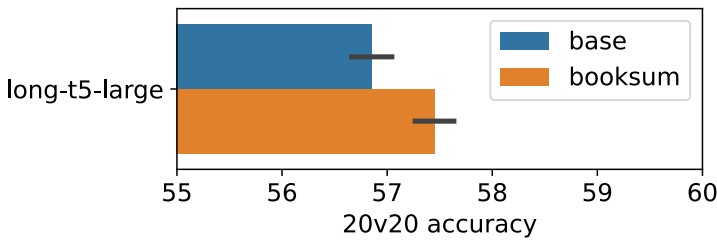

Figure 39: 20v20 brain alignment for the larger variant of LongT5, with 24 layers. We report results for both the base and booksum models of LongT5-large. Averages were computed over 8 layers for each model, sequence lengths 1 to 1000, and all 8 participants. The booksum model has greater brain alignment than the base model (paired t-test, with FDR corrected for multiple comparisons at significance level 0.05).

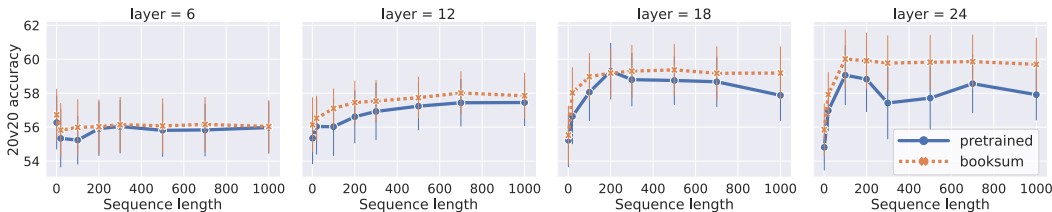

Figure 40: Comparing 20v20 brain alignment for long-t5-large-base against long-t5-large-booksum, across different layers and input sequence lengths. Each data point is an average across 8 participants. Brain alignment improvements occur in the middle and late layers (from layers 12 to 24) (paired t-test, with FDR corrected for multiple comparisons at significance level 0.05).

We ran experiments to evaluate the brain alignment for two model variants of LongT5-large with 24 layers. The first model (long-t5-large-base) was pretrained by masking and generating sentences on the C4 dataset, a collection of roughly 750 GB of English texts sourced from the public Common Crawl open repository. The second model (long-t5-large-booksum) initializes weights from the first model, and further trains the model to generate summaries for the BookSum narrative dataset. Our results show that the booksum model has greater brain alignment than the base model, and the brain alignment improvements occur in the middle and late layers (from layers 12 to 24).

In BigBird, the brain alignment improvements occur in the middle layers. On the other hand, for both the 12-layer and 24-layer variants of LongT5, the brain alignment improvements occur in the middle and late layers. For different models (BART, LED, BigBird, LongT5), we observed a different pattern of brain alignment improvements across layers and sequence lengths. We find this observation fascinating, and we hope to explore in a future work the reasons why different models improve brain alignment differently even though they are all trained for deeper understanding.

## Q  NOISE CEILING ESTIMATES FOR 20V20 METRIC

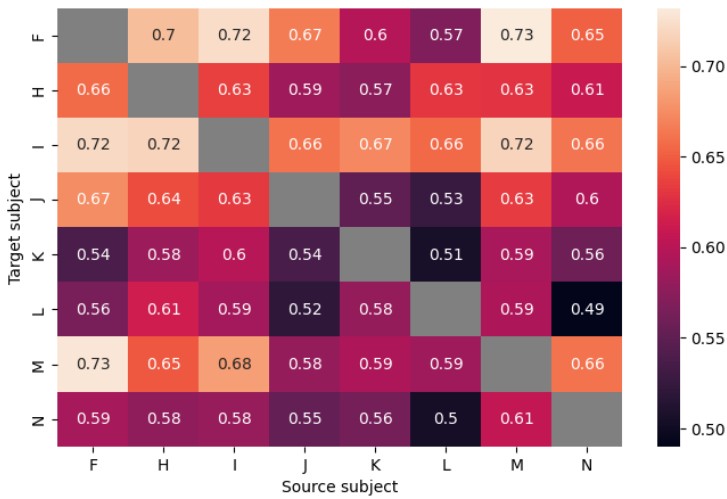

Figure 41: 20v20 results for 8*7=56 pairs of (target subject, source subject).

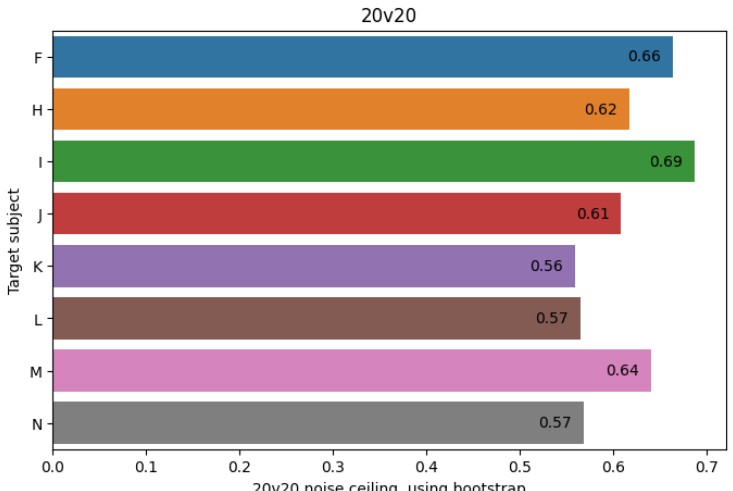

Figure 42: 20v20 Noise ceiling estimates for each of the 8 target subjects.

We ran experiments to compute the noise ceiling estimates for all 8 subjects, for the 20v20 metric. To do this, we followed previous work on computing a noise ceiling for naturalistic data with only 1 repetition of the stimulus (Schrimpf et al., 2021). We point the reader to this previous work for a detailed description of the steps. Briefly, this noise ceiling estimation is based on the ability to predict data from one participant using data from a different participant. The prediction is done by training linear encoding models that use one source subject's fMRI response to predict a different target subject's fMRI response, in a cross-validated manner.

The average noise ceiling across participants is 0.61 +/- 0.016 (sem) for 20v20. For sequence length 500, all BookSum models reach 98% of this noise ceiling (20v20 values of 0.6, for different layers across models).

