# OpenReview forum: "Training language models to summarize narratives improves brain alignment"
_ICLR.cc/2023/Conference — ICLR 2023 notable top 25%_

### Official Review · Reviewer_ajVU · 2022-10-24

**Confidence:** 4
**Correctness:** 3
**Technical Novelty And Significance:** 3
**Empirical Novelty And Significance:** 3
**Recommendation:** 6

**Clarity, Quality, Novelty And Reproducibility:**

**Clarity and Quality**: This paper is generally well written and of high quality, though the presentation could be improved. My main suggestion is to make the paper (excluding the appendices) more self-contained, by including (part of) the content of Figures 5-8 into the main content. For example, a table with numbers on a slice of sequence length would work better.

**Novelty**: The main contribution of this paper is presenting the finding that finetuning on narrative corpora improves LM-brain alignment. While models and most techniques involved in this paper has been proposed, I think the paper is of enough novelty to be considered for ICLR.

**Reproducibility**: I think the work is reproducible based on the information provided in the paper.

**Strength And Weaknesses:**


## Strengths

- This paper presents a reasonable finetuning strategy (i.e., finetuning on narrative text) to improve the alignment between LM representations and fMRI.

- Through comprehensive experiments, this paper presents fairly convincing findings on the linear alignment between pretrained language models and human brains, suggesting that further pretraining/finetuning on narrative corpora improves the alignment.

- The paper is generally clear and easy to follow.

## Weaknesses

The major weakness of this paper, in my opinion, is that it doesn't involve sufficient baselines to support the claim. For example, I would expect a baseline that finetunes the language model on non-narrative corpora, e.g., Wikipedia or scientific papers: comparing the results with those by the BookSum models in this paper would make the claims much stronger.

In addition, the following points are not clear enough to me.

- According to Wehbe et al. (2014), all 8 participants have read the Harry Potter books or watched the movies prior to the experiment. I'm not sure if the narrative models are in similar situations as well, i.e., have seen sufficient text relevant to Harry Potter in the pretraining process. If not, the story will be subtly slightly different from its current shape. I would suggest the authors to clarify this point.

- Why did the authors choose models of a different size for BigBird?
While all other investigated models have 12 layers, the selected BigBird model has 32 layers. There exists a [12-layer BigBird model](https://huggingface.co/google/bigbird-base-trivia-itc), and I wonder why the authors did not use it instead for a more uniform setting across investigated language models.

  It's somewhat a commonsense in the NLP community that the base models (named following the HuggingFace convention for models that usually have 12 layers; not to be confused with the usage of "base" in this paper) usually have different behaviors from the large models (which usually have 24 or more layers) [1, *inter alia*]. While it's very interesting to see that the alignment get worse when the the depth of layer increases, I look forward to more evidences from experiments with other deeper models (e.g., `t5-large`).

I look forward to the authors' clarification on the above points.

[1] Tenney et al., ICLR 2019. https://openreview.net/pdf?id=SJzSgnRcKX


**Summary Of The Paper:**

This paper investigates the alignment between human brains and pretrained large language models (LMs), by learning linear mappings between LM representations and fMRI data on the same corpus. In addition, the authors propose to finetune on narrative text, and find that the alignment can be improved across models by such a process.

**Summary Of The Review:**

This paper presents an intuitive finetuning strategy (i.e., finetuning on narrative text) to improve the alignment between LM representations and fMRI. Through comprehensive experiments, this paper presents fairly convincing findings on the linear alignment between pretrained language models and human brains, suggesting that further pretraining/finetuning on narrative corpora improves the alignment. There also exists some points that are not completely convincing, including lack of baselines for controlled experiment and the choices on the investigated NLP model. While I hereby give a score of 5, I am willing to raise my rating if the authors can address some or all of my concerns.

---

> ### Author Response · Authors · 2022-11-18
> **Response to Reviewer ajVU**
>
> We thank the reviewer for their thoughtful comments. We respond to their points individually below.
>
> **Q1: Additional baselines. In particular, finetuning a model on non-narrative corpora**
>
> We thank the reviewer for this important suggestion. We test whether improved brain alignment is the result of training models with a summarization objective or more specifically of training them to summarize narratives. We present the brain alignment of a model trained to summarize CNN news data in Appendix K. This model performs on par with the corresponding base pretrained model at brain alignment, suggesting that the summarization objective is not sufficient for improving brain alignment.
>
> We have further included the following additional baselines as responses to other reviewers: fine-tuned our models on the BookSum dataset with a language modeling objective (see Q1 for Z82i), computed the brain alignment for GPT-2 which is the SOTA baseline for brain alignment (see Q2 for m5W3), computed the noise ceiling measures for 20v20 which act as an estimate of the explainable variance in the brain dataset (see Q3 for m5W3), and finally controlled for an equal number of TRs in the train and test set for all discourse features (see Q4 for m5W3).
>
> **Q2: Why did the authors choose models of a different size for BigBird**
>
> We chose 4 booksum models that were among the top-performing models on the BookSum summarization benchmark. This ensures that we investigate models that all perform well at summarizing narratives (at least on the BookSum dataset). The 12-layer BigBird does not appear on the leaderboard so we opted to investigate the larger version, which is one of the best performing models.
>
> In addition, we used 3 models with 12 layers, and showed that all have improved brain alignment after training for deeper understanding. We wanted to investigate if this was also true for deeper models. We agree that evaluating additional large models would strengthen this point and have now analyzed one additional large model (see Q3 response below), which further supports our findings that models trained on the BookSum dataset have improved brain alignment.
>
> **Q3: Experiments with additional deeper models (e.g. t5-large)**
>
> We now additionally evaluate the brain alignment for LongT5-large with 24 layers. We chose this model as it is among the top-performing models on the BookSum summarization benchmark, similar to the other 4 models that we already investigated. We include the results in Appendix O. BookSum-trained LongT5-large has greater average brain alignment than its base model across layers and sequence lengths. Brain alignment improvements occur in the middle and late layers (from layers 12 to 24).
>
> We note that the layers where brain alignment improvements occur in booksum models are not always the same across the 5 models we investigate, and that these differences don’t necessarily align with the size of the model (e.g. BigBird has improvements mostly in the middle layers, whereas both LongT5 and LongT5-large have improvements in the middle and late layers). What is common across all models is that the middle layers are significantly affected, but in addition to the middle layers, some models show differences in their early layers whereas others in their late layers. We find this observation fascinating, and hope to explore the underlying reasons in future work.
>
> **Q4: Whether the pretrained models have seen information about Harry Potter before**
>
> We are not sure we fully understand the reviewer’s concern. The amount of information a pretrained model has seen about Harry Potter (HP) does not affect our conclusions about the differences in brain alignment between pretrained and BookSum models. We only make conclusions about the differences between pretrained models and the same models trained on the BookSum dataset. The BookSum dataset does not contain any information about the HP books so both the base and booksum models have seen the same information regarding HP. Thus, any brain alignment differences between the two cannot be due to the amount of information that the models have previously seen about HP. If the reviewer has something else in mind, we would appreciate it if the reviewer could elaborate on that.
>
> **Q5: Including (part of) the content of Figures 5-8 into the main content**
>
> We added Figure 1 (Left), showing the aggregated 20v20 accuracy for all 4 pairs of base and booksum models. We thank the reviewer for their suggestion, as we agree that this helps to make the paper more self-contained and easier for readers to visualize our central claim: training language models for deeper understanding improves brain alignment.
>
> We also kept Figure 1 (Right) in our paper, showing the 20v20 accuracy for BigBird, as it illustrates the pattern of brain alignment across different layers and sequence lengths. We included the same figures for the other models in the appendix due to space constraints for the main paper.

---

> > ### Comment · Reviewer_ajVU · 2022-12-12
> > **Thank you for your response!**
> >
> > Thanks for your response which resolves (almost) all my concerns!
> >
> > Regarding Q4, I agree that the point I was trying to make is vague and the difference is subtle. Here are two cases: (I) models have seen HP in pretraining; (II) models have not seen HP in pretraining. Since that the fMRI data are collected from participants who are familiar with HP, and that the authors are aligning models to human brains, it sounds a fair comparison to me if (I) is the case. If (II) is true, then aligning to brain signals of people who haven't read HP would be good.
> >
> > I hope the above clarification makes sense. Also, I'm aware that the model should have seen HP Wikipedia pages in the pretraining corpora so that the approaches presented in this paper should be good.
> >
> > I've raised my rating to 6 to show my recommendation for acceptance, and I'm actually more optimistic about the potential of this paper---I'd especially like to see more non-narrative baseline, but that's definitely not a required action item for the authors right now.

---

### Official Review · Reviewer_nyjm · 2022-10-24

**Confidence:** 3
**Correctness:** 3
**Technical Novelty And Significance:** 4
**Empirical Novelty And Significance:** 3
**Recommendation:** 8

**Clarity, Quality, Novelty And Reproducibility:**

This is a very solid paper, which appears to be novel, and has the code to replicate the experiments.
The only concern is related to the fact that fMRI is a measure that can record only brain activity in general and not at the word level. Indeed, fMRI records the blood flux in a particular region of the brain. This is used as a proxy of brain activity. Hence, a particular activity can be associated only to a large bunch of heard words.


**Strength And Weaknesses:**

PRO

- The paper is a solid experimental paper which presents an apparently novel idea

CONS

- It is difficult to interpret results if the reader is not completely aware of the field


**Summary Of The Paper:**

This paper aims to study whether particular large pre-trained transformers may align with human brain activity. This is taken as a solid hint that these pre-trained transformers are learning more than simple replicating the input. These models are domain adapted on book texts where they have to learn plots.


**Summary Of The Review:**

The ideas in the paper are really interesting and fascinating. Indeed, the parallel between machines and brains has always been used to discover new insights in computer science as well as in understanding brain activity. This parallel has also been studied in:
Zanzotto, F.M., Croce, D. (2009). Reading What Machines “Think”. Brain Informatics. 2009.
Dell’Arciprete, et al (2012). Parallels between Machine and Brain Decoding. In: Brain Informatics. 2012.

---

> ### Author Response · Authors · 2022-11-18
> **Response to Reviewer nyjm**
>
> We thank the reviewer for their thoughtful comments and positive initial evaluation. We have added additional explanations to improve our paper’s clarity. We believe this has strengthened our paper.
>
> **Q1: Make it more accessible for readers who are not completely aware of the field**
>
> We are very thankful to the reviewer for bringing up this point, as we want our work to be accessible  to readers outside the brain-NLP field. Hence, we have added explanations to clarify our thought process. (1) We added a paragraph at the end of Section 1 to discuss why we believe our results and contributions present evidence of “deeper understanding” in NLP models. (2) In Section 5, we added two paragraphs to explain our thought process behind why we varied the input context sequence length to the models, as well as what our results signify. (3) We amended the first paragraph of Section 6 to clarify how richer representations of the discourse features we selected (Characters, Emotions, Motion) suggest a deeper linguistic understanding of the text. The main reason is that these discourse features have representations that go beyond the meaning of their individual words. For example, characters develop over the course of the entire book through their thoughts, actions, and interactions with other characters. Hence, improved brain alignment for characters shows richer, more context-integrated representations which point towards better understanding of the characters.
>
> We hope that these additional explanations will make our writing more widely accessible.
>
> **Q2: fMRI BOLD signal is related to many words**
>
> As the reviewer pointed out, the response measured by fMRI is an indirect consequence of brain activity that peaks about 6 seconds after stimulus onset. To account for this, we and others who use predictive methods commonly include input features that correspond to stimuli presented during preceding timepoints (Nishimoto et al.,2011; Wehbe et al., 2014a; Huth et al., 2016). This allows for a data-driven estimation of the hemodynamic response functions (HRFs) for each voxel, which is preferable to assuming one because
> different voxels may exhibit different HRFs.
>
> The encoding models that we train take into account the delay in hemodynamic response in this way, and model each TR as a function of the words that occur in the previous 4 TRs (which corresponds to 8 seconds and 16 words). We have included this and other details about the training of the encoding model in Appendix C due to space constraints in the main paper, but we would be happy to make space for these details upon the reviewer’s request.

---

### Official Review · Reviewer_m5W3 · 2022-10-25

**Confidence:** 4
**Correctness:** 3
**Technical Novelty And Significance:** 4
**Empirical Novelty And Significance:** 4
**Recommendation:** 5

**Clarity, Quality, Novelty And Reproducibility:**

The clarity needs more work. The paper is confusing in how the two brain-alignment measures are presented. The biggest problem is that the presentations for the two metrics are pretty imbalanced. For example, the authors started in Fig. 1 with a per-layer 20vs20 performance figure. What about the same figure for the Pearson correlation metric? There should be figures showing the overall/aggregated improvement for all models to support the major claim of the authors, but I cannot find those results easily. For the 20vs20 metric, I can find some “all” measures separately in the subfigure for each model. Why don’t the authors build one figure showing all these “all” measures and put that in Fig 2 right? Fig 2 is confusing in its design: I was expecting that the right panel should show the corresponding brain measures for all models or even have two panels for each of the metrics; instead, it was just for one model, just one measure. There are also some minor inconsistencies in the writings and the figures. Like the Character increase was said to be 0.011 in the text, but the figure in the main paper only showed 0.009. After checking the Supp. Info., I found that 0.011 was for model BART.

The results seem to be reproducible. The proposed finetuning methods are also novel.


**Strength And Weaknesses:**

Strength:

This work shows that further training language models on some text corpus can improve their brain alignment, which is exciting and inspiring. This is especially interesting as the text corpus does not contain the data used in the brain data, making this improvement general. Showing that this improvement exists in both brain-alignment metrics and all four models trained makes the results general. The analysis presented by these authors is also interesting, like the discourse feature analysis. In addition, I highly appreciate the share of the code used in the paper and having many results in the supplementary information.

Weaknesses:

Two significant issues need to be further improved for this paper. First, more results are required to make the results even stronger and more general. Second, the writing of this paper is confusing and needs some work, which will be mentioned in the next part. There are also several minor issues that would be great to be addressed.

For the result issue, the biggest question is whether these four base models tested in the paper are strong candidates compared to the current SOTA brain model, like GPT-2. In both Schrimpf et al. 2021 and Goldstein et al. 2022, GPT-2 seems to be the best language model for the brain. What is the 20vs20 and Pearson correlation for this model on this dataset? Even if it may not be possible for GPT-2 to be finetuned on the book corpus, I think having the performance of that model is still important to tell us whether the improved performance leads to a new STOA.

For the minor issues, I will start with interpreting the 20vs20 metric. From my understanding, this metric can also be applied to the ground truth fMRI responses, which may lead to some noise ceiling measures for this metric. But I don’t see any discussion or results on this. Do I misunderstand something?

Then, about the results of the Character feature. Can the authors provide more measures to compare this feature group to others? For example, how many tokens are in this feature group and in other groups? Are the tokens belonging to different groups also distributed in different ways? I think this type of comparison is needed to show that the larger increase is not due to some other reasons.
Finally, I wonder whether the authors tried the same finetuning on other datasets to show the specific usage of long context in the book corpus, like showing that finetuning on some typical corpus will not work.


**Summary Of The Paper:**

This work shows that after finetuning the pretrained language models on a book corpus, the models align better with brains though their language ability is not better. This better alignment seems general to all brain regions and is also shown in two different metrics. After further evaluating the alignment increase on different discourse features, the authors state that the increase in the Characters is the largest.

**Summary Of The Review:**

This paper shows that after finetuning the pretrained language models on a book corpus, the models align better with brains though their language ability is not better. Although the results are exciting and inspiring, I think some more work (which should be doable during the rebuttal phase) is needed to make it acceptable.

---

> ### Author Response · Authors · 2022-11-18
> **Response to Reviewer m5W3**
>
> We thank the reviewer for their thoughtful comments. We respond to their points below:
>
> **Q1: Including both evaluation of brain alignment with both 20v20 accuracy and Pearson correlation metrics:**
>
> In Appendix F, we now include both metrics for the aggregate brain alignment for each of the models. We note that the trends are qualitatively similar across the two metrics. However, we observe that Pearson correlation has higher variance so we choose to use 20v20 in the main paper when possible.
>
> **Q2: Comparing brain alignment of our models against GPT-2**
>
> Upon the reviewer’s request, we ran experiments to compute brain alignment of GPT-2 in our dataset and present the results in Appendix M. For 20v20, LongT5-booksum has greater brain alignment than all other models, including GPT-2. For Pearson correlation, LongT5-booksum and GPT-2 have similar performance, and are both greater than the other models.
>
> Note that caution is needed when interpreting differences in brain alignment between models that have different base architectures and are trained on vastly different datasets. This is why in our work, we choose to focus on comparing brain alignment between pairs of models that only differ on the additional training on the Booksum dataset.
>
> **Q3: Noise ceiling measures for the 20v20 metric**
>
> We thank the reviewer for this suggestion. We ran experiments to compute the noise ceiling estimates for 20v20 for all 8 subjects. To do this, we followed previous work on computing a noise ceiling for naturalistic data with only 1 repetition of the stimulus (Schrimpf et al 2021). We include the details of this computation, and the results, in Appendix P. Briefly, the intuition behind the noise ceiling computation is to estimate the degree to which data from one participant can be predicted using data from other participants. We will make the voxel-wise values of the noise ceiling for each participant publicly available so that other researchers can easily compare their performance against them.
>
> The average noise ceiling across participants is 0.61 +/- 0.016 (sem) for 20v20. For sequence length 500, all BookSum models reach 98% of this noise ceiling (20v20 values of 0.6, for different layers across models).
>
> **Q4: Whether the larger increase for Characters is due to other reasons**
>
> There are roughly 1200 fMRI TRs in the Harry Potter dataset where predictions are generated. Among this set, there are 242 Characters TRs, 174 Emotion TRs, 165 Motion TRs, and the remaining are TRs labeled as “Others”. For each of the 3 discourse features, their TRs are relatively evenly distributed across the 1200 TRs. All 3 discourse features’ TRs are also distributed in similar ways.
>
> We agree with the reviewer that there is a possibility that the result of Characters improving more (booksum minus base) than other discourse features may be due to other reasons. The main possibility would be that there are a different number of TRs for each discourse feature in the training set (242 for Characters, 174 for Emotion, 165 for Motion). Hence, we ran additional experiments where we trained the linear encoding models on a more balanced set of TRs (the last 500 TRs of the story: 97 for Characters, 99 for Emotion, 72 for Motion). These last 500 TRs map to 2000 out of 5176 words. Among these 2000 words, there are 103 Character words, 110 Emotion words, and 69 Motion words. We then again sampled an equal number of TRs (74) for each discourse feature during testing on the first 700 TRs. We present the results in Appendix N. After controlling for an equal number of TRs in the training and test set across all discourse features, we still see the result that Characters improved more (booksum minus base) than other discourse features during training for deeper understanding.
>
> **Q5: Finetuning on non-narrative datasets**
>
> We thank the reviewer for this important suggestion. We test whether improved brain alignment is the result of training models with a summarization objective or more specifically of training them to summarize narratives. We present the brain alignment of a model trained to summarize CNN news data in Appendix K. We show that this model performs on par with the corresponding base pretrained model at brain alignment, suggesting that the summarization objective is not sufficient for improving brain alignment.
>
> **Q6: Figures showing the overall/aggregated improvement for all models**
>
> We added Figure 1 (Left) that shows the aggregated 20v20 accuracy for all 4 pairs of base and booksum models, and added Appendix F for the Pearson correlation plot. For all 4 pairs, the booksum model has significantly higher brain alignment, for both 20v20 and Pearson. We thank the reviewer for their suggestion, as we agree this makes it clearer and easier for readers to visualize our central claim: training language models for deeper understanding improves brain alignment.

---

### Official Review · Reviewer_Z82i · 2022-10-25

**Confidence:** 4
**Correctness:** 3
**Technical Novelty And Significance:** 2
**Empirical Novelty And Significance:** 4
**Recommendation:** 8

**Clarity, Quality, Novelty And Reproducibility:**

- Quality
	- There should be more controls for confounds in the training regime. As I understand it, the evaluation compares pretrained language models with the same architectures fine-tuned on another task. Two possible confounds here:
		- Effect of training environment: the mere training environment (e.g. doing gradient updates with the hyperparameters chosen by the authors) may affect brain decoding performance. The authors can control for this by fine-tuning the architectures on their original training data with a language modeling objective and with the same/similar optimization environment used for BookSum.
		- Effect of domain shift: while the training domains of the pretrained models and the Booksum models are somewhat similar, there's still a chance that the BookSum domain is what is helping the models. The authors can control for this by fine-tuning the architectures on the BookSum data (or their original training data mixed with the BookSum data) with a language modeling objective.
- Novelty: this fine-tuning-plus-brain-mapping paradigm is not entirely novel, but this use case with narrative summarization and their follow-up empirical experiments are original to the best of my knowledge.

**Strength And Weaknesses:**

- The paper addresses a timely question about the sorts of language training objectives which produce brain-like representations. This question is of interest to several different fields within cognitive science/AI.
	- Note that this is not exactly the framing the authors themselves choose: they seem to believe that brain mapping performance can select between different types of models in deciding which models are "deep" in the same sense that human language processors are. I'm skeptical on this, but I think the methods and results have value in any case.
- Good effort at interpreting the results with several creative follow-up experiments (sec. 5: is improved performance reducible to language modeling?; sec. 6: what sorts of representational content are driving improved performance?).
- The notion of "deeper understanding" is never clearly defined *a priori* in the text, except as something that is instantiated in the BookSum task, and something which isn't reducible to language modeling.
	- A posteriori, it also seems to be something which yields model representations with improved brain encoding performance—across all tested ROIs!
	- For a paper at this conference, however, I don't think this is a critical issue. I'd encourage the authors to follow up this research and test their concept of "deeper understanding" by considering alternative tasks other than narrative summarization.

**Summary Of The Paper:**

This paper presents a neural network brain encoding analysis comparing the performance of Transformer architectures trained on language modeling objectives versus those trained on a narrative summarization objective. The authors argue that the narrative summarization task is a proxy for "deeper understanding," and that the improved brain encoding performance of models optimized on the task (AKA BookSum models) demonstrates this point. They show that the improved brain encoding performance of BookSum models is not reducible to improved language modeling performance, and use a data-driven evaluation to further study the representational content that could be driving increased brain encoding performance.

**Summary Of The Review:**

Overall I think the paper does a nice job of pairing performance gains with multiple attempts at explaining their cause. I'm not especially convinced by the final interpretability analysis, but I think these results are interesting and can help to start a good discussion in the community. I vote to accept, and encourage the authors to add the controls mentioned in the "Quality" section above.

---

> ### Author Response · Authors · 2022-11-18
> **Response to Reviewer Z82i**
>
> We thank the reviewer for their thoughtful comments and positive initial evaluation. We respond to their points below:
>
> **Q1: Better controlling for possible effect of domain shift: fine-tuning architectures on the BookSum data with a language modeling objective**
>
> We now include brain alignment results for models fine-tuned on the BookSum data with a language modeling objective (referred to as “LM-booksum”) in Appendix L. For 3 of the 4 models, LM-booksum models outperform the base models, which suggests that the domain shift does play a role in the brain alignment. Still, for 2 of the 4 models, the booksum models outperform the corresponding LM-booksum model at brain alignment, which shows that the improved brain alignment for the booksum models is not simply due to domain similarity. For the remaining 2 models, booksum and LM-booksum perform on par at brain alignment.
>
> Note that the LM-booksum models have substantially better LM performance on the fMRI dataset text than the booksum models (see Appendix L). As LM performance has been shown to contribute to brain alignment (Schrimpf et al. 2021, Goldstein et al. 2022), it is difficult to disentangle the brain alignment contributions of LM performance and similar narrative domain for the LM-booksum models. The LM-booksum models have greater LM performance than both the base and booksum models, whereas the base and booksum models have similar LM performance. Hence, we believe that the LM-booksum models may not be a better general baseline than the base models.
>
> **Q2: Better controlling for effect of training regime and environment**
>
> The respective pretraining datasets are very large and training on them with different hyperparameters is not feasible for us during this response period. As an alternative, we tried using similar optimization environments for LM-booksum as the ones used for the booksum models. These included training on the same dataset, using long input sequences (above 1024) for training, and using similar optimization hyperparameters and the Adam optimizer. We hope that this addresses the reviewer’s concern sufficiently.
>
> Additionally, as a response to a suggestion by reviewer ajVU (see Q1) and m5W3 (see Q5), we also test whether improved brain alignment is the result of training models with a summarization objective or more specifically of training them to summarize narratives. We present the brain alignment of a model trained to summarize CNN news data in Appendix K. We show that this model performs on par with the corresponding base pretrained model at brain alignment, suggesting that the summarization objective is not sufficient for improving brain alignment.
>
> **Q3: Elaborating how brain mapping performance can be used to suggest deep understanding in NLP models, and defining the notion of “deeper understanding”**
>
> We added a paragraph at the end of Section 1 to discuss how our results and contributions present evidence of “deeper understanding” in NLP models. There are 3 main arguments. First, improved alignment to a human brain’s deep understanding of characters, emotions and motions
> suggests that the model has developed richer representations of these entities and concepts. Second, we specifically focus on brain regions that previous research has shown underlie language comprehension in humans. Hence, improved brain alignment is not spuriously related to non-language brain activities. Third, we show that brain alignment does not improve when we only provide short input contexts (1 to 5 words) to the LMs. It improves only for longer contexts (20 to 500 words), which is more likely where deep contextual understanding is needed.
>
> We have also added a paragraph at the start of Section 6 to clarify the notion of “deeper understanding”. Characters, emotions and motions in a narrative can have representations that go beyond the meaning of their individual words. For example, characters develop over the course of the entire book through their thoughts, actions, and interactions with other characters. Hence, we interpret improved brain alignment for characters to show richer, more context-integrated representations which point towards better understanding of the characters. We agree that an exciting future direction is to test the concept of “deeper understanding” by considering alternative tasks other than narrative summarization.
>
> **Q4: For the discourse interpretability analysis, we provide additional experimental controls and improved explanations**
>
> As a response to Reviewer m5W3 (see Q4), we provide an additional control analysis for the improvement that is related to specific discourse features (see Appendix N). In this control analysis, we not only balance the number of samples of each discourse feature type in the test set, but also in the training set. We show that our previous results that brain alignment improves more for samples that contain Characters than for other discourse features hold also in this analysis.

---

### Author Response · Authors · 2022-11-18
**Common response to all reviewers**

We thank all reviewers for their helpful feedback and for thoroughly evaluating our work. We have attempted to address their concerns and incorporate their suggestions. We believe that this has strengthened the work. Here we summarize the additional analyses that we provide to strengthen our claims, and respond to each reviewer individually in the threads below.

1. brain alignment of model trained to summarize non-narrative text (CNN articles) (see Q1 for Reviewer ajVU and Q5 for Reviewer m5W3)
2. brain alignment for fine-tuned models on the BookSum dataset with a language modeling objective (see Q1 for Reviewer Z82i)
3. comparisons with brain alignment for GPT-2 which is the SOTA brain alignment baseline (see Q2 for Reviewer m5W3)
4. computed the noise ceiling measures which can be thought of as an estimate of the explainable variance in the brain dataset, and committed to making them publicly available since this is a popular brain dataset (see Q3 for Reviewer m5W3)
5. controlled for an equal number of TRs in the training and test set for all discourse features in the interpretability analysis (see Q4 for Reviewer m5W3)
6. brain alignment for one additional deeper model before and after training with the summarization objective on the BookSum dataset (long-t5-large, 24 layers) (see Q3 for Reviewer ajVU)

We hope that these additional analyses and our individual responses will alleviate many of the reviewers’ concerns and are looking forward to engaging with the reviewers in the case that there are remaining concerns.

---

### Author Response · Authors · 2022-12-06
**Checking in with reviewers**

As the end of the reviewing period is approaching, we would like to ask the reviewers whether the 6 added analyses that we describe below have sufficiently addressed their main concerns or if there are any outstanding questions. We really appreciate the reviewers’ feedback and suggestions and believe that our work has been strengthened by the added analyses.

---

### Decision · Program_Chairs · 2023-01-20

**Decision:**

Accept: notable-top-25%

**Justification For Why Not Higher Score:**

See weaknesses in meta-review

**Justification For Why Not Lower Score:**

See strengths in meta-review

**Metareview: Summary, Strengths And Weaknesses:**

This paper investigates the alignment between human brains and pretrained large language models (LMs), by learning linear mappings between LM representations and fMRI data on the same corpus. In addition, the authors propose to finetune on narrative text, and find that the alignment can be improved across models by such a process.

**Strengths:**
All authors found the experimental setting of measuring alignment between PLM and brain representations interesting. There was also excitement about the finding that finetuning on narrative corpora (i.e., stories) yielded even more aligned representations. Certain reviewers would have preferred to see another finetuned baseline, so the authors added such an experiment during the rebuttal demonstrating that training on summarization data did not provide the same improvements.

**Weaknesses:**
Certain reviewers felt the paper could have been written more clearly.

I think this is an interesting area of inquiry. The authors have formulated a thorough and substantial finding on LM-brain alignment that I think should be presented at the conference.


**Note From Pc:**

if the above contains the word "oral" or "spotlight" please see: "oral" presentation means -> notable-top-5% and "spotlight" means -> notable-top-25%. As stated in our emails, we are disassociating presentation type from AC recommendations

**Summary Of Ac-Reviewer Meeting:**

N/A